



# Potential evaporation at eddy-covariance sites across the globe

Wouter H. Maes[1], Pierre Gentine[2], Niko E.C. Verhoest[1], Diego G. Miralles[1]

1 Laboratory of Hydrology and Water Management, Ghent University, Coupure Links 653, 9000 Gent, Belgium
Department of Earth and Environmental Engineering, Columbia University, New York, 10027, USA

*Correspondence to:* Wouter H. Maes (wh.maes@ugent.be)

**Abstract.** Potential evaporation ($E_p$) is a crucial variable for hydrological forecasting and drought monitoring. However, multiple interpretations of $E_p$ exist, and these reflect a diverse range of methods to calculate it. As such, a comparison of the performance of these methods against field observations in different global ecosystems is urgently needed. In this study, potential evaporation was defined as the rate of evaporation (or evapotranspiration - sum of transpiration and soil

evaporation) that the actual ecosystem would attain if it evaporates at maximal rate. We use eddy-covariance measurements from the FLUXNET2015 database, covering eleven different biomes, to parameterize and inter-compare the most widely used $E_p$ methods and to uncover their relative performance. For each site, we isolate the days for which ecosystems can be considered as unstressed based on both an energy balance approach and a soil water content approach. Evaporation measurements during these days are used as reference to calibrate and validate the different methods to

estimate $E_p$. Our results indicate that a simple radiation-driven method calibrated per biome consistently performs best, with a mean correlation of 0.93, unbiased RMSE of 0.56 mm day$^{-1}$, and bias of -0.02 mm day$^{-1}$ against *in situ* measurements of unstressed evaporation. A Priestley and Taylor method, calibrated per biome, performed just slightly worse, yet substantially and consistently better than more complex Penman, Penman-Monteith-based or temperature-driven approaches. We show that the poor performance of Penman-Monteith-based approaches relates largely to the fact

that the unstressed stomatal conductance cannot be assumed to be constant in time at the ecosystem scale. Contrastingly, the biome-specific parameters required for the simple radiation-driven methods are relatively constant in time and per biome type. This makes these methods a robust way to estimate $E_p$ and a suitable tool to investigate the impact of water use and demand, drought severity and biome productivity.

*(insert abbreviation list)*

## 1 Introduction

Since its introduction 70 years ago by C. W. Thornthwaite (1948), the concept of potential evaporation ($E_p$), defined as the amount of water which would evaporate from a surface unconstrained by water availability, has been widely used in multiple fields. It has been incorporated in hydrological models dedicated to estimate runoff (e.g. Schellekens et al., 2017)

or actual evaporation (Wang and Dickinson, 2012) as well as in drought severity indices (Sheffield et al., 2012;Vicente-Serrano et al., 2013). Changes in $E_p$ have been regarded as a driver of ecosystem distribution and aridity (Scheff and





Frierson, 2013) and used to estimate the influence of climate change on ecosystems based on climate model projections (e.g. Milly and Dunne, 2016).

However, many different definitions of $E_p$ exist, and consequently many different methods are available to calculate it. In recent years, there has been an increasing awareness of the impact of the underlying assumptions and caveats in

traditional $E_p$ methods (Weiß and Menzel, 2008;Kingston et al., 2009;Sheffield et al., 2012;Seiller and Anctil, 2016;Bai et al., 2016;Milly and Dunne, 2016;Guo et al., 2017). Consequently, a global appraisal of the most appropriate method for assessing the $E_p$ of actual ecosystems is urgently needed. Yet, current methods disagree on the mere meaning of this variable, which requires the definition of a reference system (Lhomme 1997). $E_p$ has been typically defined as the evaporation which would occur in given meteorological conditions if water was not limited, either (*i*) over open water

(Shuttleworth, 1993); (*ii*) over a reference crop - usually a wet (Penman, 1963) or irrigated (Allen et al., 1998) short green grass completely shading the ground - or (*iii*) over the actual ecosystem transpiring at a maximal rate (Brutsaert, 1982;Granger, 1989).

A second source of disagreement on the definition of $E_p$ relates to the spatial extent of the reference system and the consideration (or not) of feedbacks from the reference system on the atmospheric conditions. Several authors found it

convenient to define $E_p$ taking an extensive area as a reference system, because considering this reduces the influence of advection and entrainment flows (Penman, 1963;Priestley and Taylor, 1972;Brutsaert, 1982;Shuttleworth, 1993). Such an idealized extensive and well-watered ecosystem evaporating at maximum rate can be expected to raise air humidity until the vapour pressure deficit tends to zero. If this feedback of the extensive system on the aerodynamic forcing (effect of VPD and wind speed) is considered, evaporation will be driven by radiation only. Meanwhile, others have defended

the use of reference systems that are infinitesimally small (Morton, 1983;Pettijohn and Salvucci, 2009;Gentine et al., 2011b), in order to avoid this feedback of the reference system on the aerodynamic forcing in the calculation $E_p$. The effect of this choice of reference system is best exemplified by the complementary relationship framework (Bouchet, 1964), which uses both approaches to link potential and actual evaporation, through $(1 + b)\,E_{p0} = E_{pa} + b\,E_a$, with $b$ an empirical constant (Kahler and Brutsaert, 2006;Aminzadeh et al., 2016), $E_{p0}$ the evaporation from an extensive well-

watered surface (i.e. in which the feedback from the ecosystem on the VPD and aerodynamic forcing is considered and where evaporation is only driven by a radiative forcing), $E_{pa}$ the evaporation from a well-watered but infinitesimally small surface (i.e. where evaporation is driven by both a radiative and an aerodynamic forcing) and $E_a$ the actual evaporation, (Morton, 1983).

Upon all this controversy, the net radiation of the reference system remains another point of discussion: some authors

argue that the (well-watered) reference system should have the same net irradiance as the actual (water-limited) system (e.g. Granger, 1989;Rind et al., 1990;Crago and Crowley, 2005). Yet, this is inherently inconsistent as the surface temperature reflects the surface energy partitioning, thus a well-watered system transpiring at potential rate is expected to have a lower surface temperature (Maes and Steppe, 2012), and correspondingly a higher net radiation (e.g. Lhomme, 1997;Lhomme and Guilioni, 2006). Meanwhile, to some extent, the albedo also depends on soil moisture (Eltahir,

1998;Roerink et al., 2000;Teuling and Seneviratne, 2008) and some argue it should ideally be adjusted to reflect well-watered conditions (Shuttleworth, 1993). Finally, extensive reference surfaces can be expected to not only exert a feedback on the aerodynamic forcing, but also on the radiative forcing by altering temperature, humidity and cloud formation. Yet, these larger-scale feedbacks are not acknowledged when computing $E_p$, even when considering extensive reference systems.

As can be concluded from the above discussion, it is nearly impossible to define a unique and universally accepted definition of $E_p$, and the most appropriate definition remains tied to the specific interest and application. Nonetheless, as different applications make use of different $E_p$ methods, a good knowledge of the implications of different $E_p$ definitions





is required (Fisher et al., 2011). When considering terrestrial ecosystems, the open water reference system seems less informative of the available energy and the aerodynamic properties of the actual ecosystem (Shuttleworth, 1993;Lhomme, 1997). The approach of considering an idealised well-watered crop system only takes climate forcing conditions into account, and requires no information on the actual land cover, which is why it has become the standard to estimate global

atmospheric demands for water or drought severity (Dai, 2011). When the actual ecosystem transpiring at maximal rate is considered the reference system, both climate forcing conditions and ecosystem properties need to be taken into account. This has been the preferred approach when calculating $E_p$ as an intermediate step to estimate actual evaporation. This is commonly done by applying a multiplicative stress function $S$ varying between 0 and 1, such that $E_a = S\,E_p$ (e.g. Mu et al., 2007;Fisher et al., 2008;Miralles et al., 2011;Martens et al., 2017). This $S$ factor can be considered analogous

to the $\beta$ factor used in some land surface models to incorporate the effect of soil moisture in the estimation of surface turbulent fluxes (Powell et al., 2013), with the difference that $S$ can include other factors than just soil moisture.

Several previous studies have attempted to compare and evaluate different $E_p$ methods. These studies have either compared the performance of different $E_p$ methods in hydrological models (Xu and Singh, 2002;Oudin et al., 2005a;Kay and Davies, 2008;Seiller and Anctil, 2016) or in climate models (Weiß and Menzel, 2008;Lofgren et al., 2011;Milly and

Dunne, 2016). Other studies considered the Penman-Monteith method as the benchmark to test less input-demanding formulations (e.g. Chen et al., 2005;Sentelhas et al., 2010). All these studies have their own merits, yet an evaluation of $E_p$ methods based on empirical data of actual measurements of evaporation is to be preferred (Lhomme, 1997). To date, such approaches have been hampered by limited data availability (Weiß and Menzel, 2008). Lysimeters provide arguably the most precise evaporation measurements available (e.g. Abtew, 1996;Pereira and Pruitt, 2004;Yoder et al., 2005;Katerji

and Rana, 2011), but they are sparsely distributed and not always representative of larger ecosystems. Pan evaporation measurements are more easily taken and broadly available (Zhou et al., 2006;Donohue et al., 2010;McVicar et al., 2012) but provide a proxy of open-water evaporation, rather than actual ecosystem potential evaporation; they also exhibit biases related to the location, shape and composition of the instrument (Pettijohn and Salvucci, 2009). Eddy-covariance measurements are an attractive alternative, but, apart from an unpublished study by Palmer et al. (2012), have so far only

been used in $E_p$ studies focusing on local to regional scales only (Jacobs et al., 2004;Sumner and Jacobs, 2005;Douglas et al., 2009;Li et al., 2016).

The overall purpose of the present work is to identify the most suitable method to estimate $E_p$ at the ecosystem-scale across the globe. Because we are using an empirical dataset of actual evaporation at FLUXNET sites, the reference system considered in this study is the actual ecosystem transpiring at maximal rate, so $E_p$ is defined as the evaporation of the

actual ecosystem when it is completely unstressed. As mentioned above, this definition is the most suitable for hydrological studies, or to study either the $S$ (or $\beta$) factor if $E_a$ is available or, instead, to calculate $E_a$ if a model for the $S$ factor is at hand. $E_p$ in this definition is similar to $E_{p0}$ in the complementary relationship. We used the most recent and complete eddy-covariance database available, i.e. the FLUXNET2015 archive (http://fluxnet.fluxdata.org/). The most frequently-adopted $E_p$ methods are applied based on standard parameterizations as well as calibrated parameters by biome,

and inter-compared in order to gain insights into the most adequate means to estimate $E_p$ from ecosystem to global scales.

## 2. Material and Methods

### 2.1. Selection of $E_p$ methods

Methods to calculate $E_p$ can be categorized based on the amount and type of input data required. In this overview, we will only discuss the ones evaluated in our study, which are arguably the most frequently used. Readers are referred to Oudin

et al. (2005a) or Seiller and Anctil (2016) for more inclusive overviews.





**Methods based on radiation, temperature, wind speed and vapour pressure**

The well-known Penman-Monteith equation (Monteith, 1965) expresses latent heat flux $\lambda E_a$ (W m$^{-2}$) as:

$$\lambda E_a = \frac{s\,(R_n-G)+\frac{\rho_a\,c_p VPD}{r_{aH}}}{s+\gamma+\gamma\frac{r_c}{r_{aH}}} = \frac{s\,(R_n-G)+\frac{\rho_a\,c_p VPD}{r_{aH}}}{s+\gamma+\frac{\gamma}{g_c\,r_{aH}}} \qquad (1)$$

With $\lambda$ the latent heat of vaporisation (J kg$^{-1}$), $E_a$ the actual evaporation (kg m$^{-2}$ s$^{-1}$), $s$ the slope of the Clausius-Clapeyron curve relating air temperature with the saturation vapour pressure (Pa K$^{-1}$), $R_n$ the net radiation (W m$^{-2}$), G the ground

heat flux (W m$^{-2}$), $\rho_a$ the air density (kg m$^{-3}$), $\gamma$ the psychrometric constant (Pa K$^{-1}$), $c_p$ the specific heat capacity of the air (J kg$^{-1}$ K$^{-1}$), VPD the vapour pressure deficit (Pa), $r_{aH}$ the resistance of heat transfer to air (s m$^{-1}$), $r_c$ the canopy resistance of water transfer (s m$^{-1}$) and $g_c$ the canopy conductance to water transfer (m s$^{-1}$; $g_c = r_c^{-1}$). While $\lambda$, $c_p$, $s$ and $\gamma$ are air temperature-dependent, $r_{aH}$ is a complex function of wind speed, vegetation characteristics and the stability of the lower atmosphere (see Section 2.3). In most methods to estimate $E_a$ or $E_p$, $r_{aH}$ is estimated from a simple function of wind speed.

The Penman-Monteith equation can be used to calculate $E_p$ by adjusting $r_c$ to its minimum value (the value under unstressed conditions). If the reference system is the actual system, or a reference crop transpiring at maximal rate, $r_c$ is usually considered as a fixed, constant value larger than zero (even if the soil is well watered). In this study, both a universal, fixed value of $r_c$ (or $g_c$) for reference crops as a biome-specific constant value are used (see Sect. 2.5). When instead of a well-watered soil, a wet canopy (i.e. a canopy covered by water) is considered, $r_c$=0 thus Eq. (1) collapses to:

$$\lambda E_p = \frac{s\,(R_n-G)+\frac{\rho_a\,c_p VPD}{r_{aH}}}{s+\gamma} \qquad (2)$$

Eq. (2) is often referred to as the Penman (1948) formulation, and can be conveniently rearranged as $\lambda E_p = \frac{s\,(R_n-G)}{s+\gamma} + \frac{\rho_a\,c_p VPD}{(s+\gamma)\,r_{aH}}$ to illustrate that $E_p$ can be driven by a radiative (left term) or aerodynamic (right term) forcing (Brutsaert and Stricker, 1979).

**Methods based on radiation and temperature**

When the reference system is considered an idealized extensive area, or when radiative forcing is very dominant, the

aerodynamic component of Eq. (2) becomes negligible and the whole equation collapses to $\lambda E_p = \frac{s\,(R_n-G)}{s+\gamma}$, which is commonly referred to as 'equilibrium evaporation' (Slatyer and McIlroy, 1961). Priestley and Taylor (1972) analysed time series of open water and water-saturated crops and grasslands and found that the evaporation over these surfaces closely matched the equilibrium evaporation corrected by a multiplication factor, which is commonly denoted as $\alpha_{PT}$:

$$\lambda E_p = \alpha_{PT}\frac{s\,(R_n-G)}{s+\gamma} \qquad (3)$$

This formulation is known as the Priestley and Taylor equation, and because usually a constant value of $\alpha_{PT}$ is adopted,

it assumes that the aerodynamic term in the Penman equation (eq. 2) is a constant fraction of the radiative term. Typically, $\alpha_{PT}$ =1.26 is considered, as estimated by Priestley and Taylor (1972) in their original experiments. In this study, we also include a vegetation-specific value to extend its applicability to all biomes (see Sect 2.5). Since this method does not require wind speed or VPD as input, it is one of the $E_p$ methods that is most widely used in hydrological models (Norman et al., 1995;Castellvi et al., 2001;Agam et al., 2010), remote sensing evaporation models (Norman et al., 1995;Fisher et

al., 2008;Agam et al., 2010;Miralles et al., 2011;Martens et al., 2017) and drought monitoring systems (Anderson et al., 1997).





**Methods based on radiation**

Other studies such as Lofgren et al. (2011), or the more recent Milly and Dunne (2016), further simplified Eq. (3) to make it a linear function of the available energy by defining a constant multiplier here referred to as $\alpha_{MD}$:

$$\lambda E_p = \alpha_{MD} (R_n - G) \tag{4}$$

In the case of Milly and Dunne (2016) this equation was applied to climate model outputs based on a constant and

universal value of $\alpha_{MD}$=0.8. The above Eq. (4) can be easily related to the surface energy balance of the ecosystem, which is given by $R_n - G = \lambda E_a + H$, with H the sensible heat flux (W m$^{-2}$). On a daily scale, $(R_n - G)$ expresses the total amount of energy available for evaporation, and the fraction of this energy that is actually used for evaporation is typically referred to 'evaporative fraction', or $EF = \frac{\lambda E_a}{(H + \lambda E_a)} = \frac{\lambda E_a}{(R_n - G)}$. From Eq. (4), it follows that the parameter $\alpha_{MD}$ can be interpreted as the EF of the unstressed ecosystem. In this study, we test both the general value of $\alpha_{MD}$=0.8 and a biome-

specific value (see Sect. 2.5).

**Methods based on temperature**

Of the many empirical methods to estimate $E_p$, temperature-based methods have arguably been most commonly used because of the availability of reliable air temperature data. For an overview of these methods, we refer to Oudin et al. (2005a). In this study, two methods are included. Pereira and Pruitt (2004) formulated a daily version of the well-known

Thornthwaite (1948) equation:

$$T_{eff}<0 \qquad \lambda E_p = 0 \tag{5a}$$

$$0<T_{eff}<26 \qquad \lambda E_p = \alpha_{Th} \left(\frac{10\,T_{eff}}{I}\right)^b \left(\frac{N}{360}\right) \tag{5b}$$

$$26<T_{eff} \qquad \lambda E_p = -c + d\,T_{eff} - e\,T_{eff}^2 \tag{5c}$$

with $T_{eff}$ being the effective temperature, based on maximum and minimal temperatures (see further, Section 2.5), $\alpha_{Th}$ an empirical parameter (see below), I the yearly sum of $(T_{a\_mean}/5)^{1.514}$, with $T_{a\_mean}$ the mean air temperature for each month, $N$ the number of daylight hours, $b$ a parameter depending on I and $c$, $d$ and $e$ empirical constants (see Section 2.5). The general value of $\alpha_{Th}$=16 is often adopted; in this study, we will also calculate and apply a biome-specific value.

The second temperature-based method is the one proposed by Oudin et al. (2005a), after comparing 27 physically-based and empirical methods with runoff data from 308 catchments:

$$T_a<5 \qquad \lambda E_p = 0 \tag{6a}$$

$$T_a>5 \qquad \lambda E_p = \frac{R_e}{\rho_a} \frac{(T_a - 5)}{\alpha_{Ou}} \tag{6b}$$

with $T_a$ the air temperature (°C), and $R_e$ top-of-atmosphere radiation (MJ m$^{-2}$ day$^{-1}$), depending on latitude and Julian day. Oudin et al. (2005a) suggested to use $\alpha_{Ou}$ = 100. This value will be used, in addition to a biome-specific value. A detailed description of the calibration of all $E_p$ methods is given in Section 2.5.

**2.2. FLUXNET2015 Database**

The Tier2 FLUXNET2015 database based on half-hourly or hourly measurements from eddy-covariance sensors is used to evaluate the estimates of $E_p$ (http://fluxnet.fluxdata.org/data/fluxnet2015-dataset/). Sites lacking at least one of the





basic measurements required for our analysis (i.e. $R_n$, G, $\lambda E_a$, H, wind speed ($u$), friction velocity ($u*$), $T_a$ and relative humidity (RH) or VPD) were not further considered. For latent and sensible heat flux, we used the data corrected by the Bowen ratio method. In this approach, the Bowen ratio is assumed to be correct, and the measured $\lambda E_a$ and H are multiplied by a correction factor derived from a moving window method; see http://fluxnet.fluxdata.org/data/fluxnet2015-

dataset/data-processing/ for a detailed description of this standard procedure. Taking the uncorrected $\lambda E_a$ instead did not impact the main findings. For the main heat fluxes (G, H, $\lambda E_a$), medium and poor gap-filled data were masked out according to the flags provided by FLUXNET. As no quality flag was available for $R_n$ measurements, the flag of the shortwave incoming radiation was used instead. All negative values for H or $\lambda E_a$ were masked out, as these relate to periods of interception loss and condensation when accurate measurements are not guaranteed (Mizutani et al., 1997).

Similarly, all negative values of $R_n$ were masked out.

Finally, sub-daily measurements were aggregated to daytime composites based on a threshold of 5 W m$^{-2}$ of top-of-atmosphere incoming shortwave radiation and the first and last (half-) hours of the day were excluded from these aggregates. Based on these daytime values, the daytime means of $s$, $\gamma$, $\rho_a$ were calculated using the parameterisation procedure described in Allen et al. (1998). We used air temperature to calculate $s$. Only days in which more than 30% of

the data were measured directly were retained, and days with rainfall (between midnight and sunset) were removed from the analyses to avoid the effects of rainfall interception. Only sites with at least 80 retained days were used for the further analysis. The global distribution of the final selection of sites is shown in Fig. 1 and detailed information about these sites is provided in Table S1 of the Supporting Information. The IGBP-classification was used to assign a biome to each site.

*(Insert Figure 1)*

### 2.3. Calculation of resistance parameters

Estimates of $r_{aH}$ are required for the Penman and Penman-Monteith equations. The resistance of heat transfer to air, $r_{aH}$, was calculated as:

$$r_{aH} = \frac{u}{u_*^2} + \frac{1}{k\,u_*} \left[ \ln\left(\frac{z_{0m}}{z_{0h}}\right) + \Psi_m\left(\frac{z-d}{L}\right) - \Psi_m\left(\frac{z_{0h}}{L}\right) - \Psi_h\left(\frac{z-d}{L}\right) + \Psi_h\left(\frac{z_{0h}}{L}\right) \right] \tag{7}$$

in which $k$=0.41 is the von Karman constant, $z$ the (wind) sensor height (m), $d$ the zero displacement height (m), $z_{0m}$ and

$z_{0h}$ the roughness lengths for momentum and sensible heat transfer (m), respectively, L the Obukhov length (m), and $\Psi_m(X)$ and $\Psi_h(X)$ the Businger-Dyer stability functions for momentum and heat for the variable X, respectively. These were calculated based on the equations given by Garratt (1992) and Li et al. (2017) for stable, neutral and unstable conditions. Note that in neutral and stable conditions, $\Psi_m(X) = \Psi_h(X)$ and that $\Psi_m\left(\frac{z-d}{L}\right) - \Psi_m\left(\frac{z_{0h}}{L}\right) - \Psi_h\left(\frac{z-d}{L}\right) + \Psi_h\left(\frac{z_{0h}}{L}\right) = 0$. This is not the case for the unstable conditions that usually prevail during the daytime. Daytime averages

of all variables were used as input in Eq. (7).

The sensor height $z$ was collected individually for each tower through online and literature research, or personal communication with the towers' P.I. The Obukhov length L was calculated as:

$$L = \frac{-u_*^3 \rho_a T_a (1 + 0.61\,q_a)\,c_p}{k\,g\,H} \tag{8}$$

with $q_a$ being the specific humidity (kg kg$^{-1}$) and $g$=9.81 m s$^{-2}$ the gravitational acceleration.





The displacement height $d$ and the roughness length for momentum flux $z_{0m}$ were estimated as a function of the canopy vegetation height (VH), as $d=0.66$ VH and $z_{0m}=0.1$ VH (Brutsaert, 1982). VH was estimated from the flux tower measurements using the approach of Pennypacker and Baldocchi (2016):

$$VH = \frac{z}{0.66 + 0.1 \exp\left(\frac{k\,u}{u_*}\right)} \tag{9}$$

This equation was applied to the full (half-)hourly database, and only when conditions were near-neutral ($|z/L| < 0.01$) and friction velocities lower than one standard deviation below the mean $u_*$ at each site. The daily VH was then aggregated by averaging out the half-hourly estimates to daily values, excluding the 20% outliers of the data, and then calculating a 30-day window moving average on the dataset. When not enough (half-)hourly vegetation height observations (<160) were available, the site was excluded from the analysis. This gave robust results for all remaining sites; an example of VH temporal development for a specific location is given in Fig. 2a. For homogeneous canopies, the VH calculated this way represents the true vegetation height. For savannah-like ecosystems, it corresponds to the vegetation height that the vegetation would have if it was represented by a single big leaf model.

The Stanton number (defined as $kB^{-1} = \ln(z_{0M}/z_{0H})$) was calculated by assuming that the surface aerodynamic temperature $T_0$ (defined by $H = \rho_a\, c_p \frac{(T_0 - T_a)}{r_{aH}}$) is equal to the radiative surface temperature $T_s$ derived from the longwave fluxes. Then, through an iterative approach, an optimal value of $z_{0H}$ was obtained, using the following equations for $T_0$ (Garratt, 1992) and $T_s$ (Maes and Steppe, 2012):

$$T_0 = T_a + \left(\frac{H}{k\,u_*\,\rho_a\,c_p}\right)\left[\ln\left(\frac{z - d}{z_{0h}}\right) - \Psi_h\left(\frac{z - d}{L}\right) + \Psi_h\left(\frac{z_{0h}}{L}\right)\right] \tag{10}$$

$$T_s = \sqrt[4]{\frac{LW_{out} - (1 - \varepsilon)\,LW_{in}}{\sigma\,\varepsilon}} \tag{11}$$

with $\sigma$ the Stefan-Boltzmann constant and $\varepsilon$ the emissivity (see further). The (half-)hourly data were used for this calculation. Following the approach of Li et al. (2017), only summertime data were used, and only when H was larger than 20 W m$^{-2}$ and $u_*$ larger than 0.01 m s$^{-1}$. Summertime was defined as those months in which the maximal daily value for H is at least 85% of the maximum value for H for the time series at the tower – with the maximum value derived as the 98th percentile, to avoid influences from outliers. In addition, (half-)hourly observations with counter-gradient heat fluxes were excluded from the analysis. For each observation, $z_{0H}$ was optimized by minimizing the difference between $T_0$ and $T_s$. Then, $kB^{-1}$ was calculated at each site based on its relation with the observed Reynolds number (Re) by fitting the following function, based on the work by Li et al. (2017):

$$k\,B^{-1}=a_0 + a_1\,Re^{a2} \tag{12}$$

Note that Eq. (11) requires a value for $\varepsilon$, which is often assumed to be equal to 0.98 for all sites (e.g. Li et al., 2017; Rigden and Salvucci, 2015). Under the assumption that $T_0=T_s$, $\varepsilon$ can also be calculated separately per site. If H=0, it follows that $T_0=T_a$ and from Eq. (11),

$$\varepsilon = \frac{LW_{out} - LW_{in}}{\sigma\,T_a^4 - LW_{in}} \tag{13}$$

Here, $\varepsilon$ was calculated for each site using (half-)hourly data, selecting those measurements where H was close to 0 (-2<H<2 W m$^{-2}$) and excluding rainy days as well as measurements in which the albedo (calculated as $\alpha = SW_{out}/SW_{in}$ with $SW_{in}$ the incoming and $SW_{out}$ the outgoing shortwave radiation) was above 0.4, to avoid influences of snow or ice.





Negative estimates of ε were masked out, and the ε of the site was calculated as the mean excluding the outlying 20% of the data. Equation 3 was applied both with a fixed ε of 0.98 and with the observed ε, and the equation with the lowest RMSE for Eq. (12) was retained. An example of such a function between $kB^{-1}$ and Re is shown in Fig. 2b.

*(Insert Figure 2)*

Finally, the canopy resistance $r_c$ (s m$^{-1}$) was calculated from the Penman-Monteith equation as:

$$r_c = \frac{s\,(R_n - G)\,r_{aH} + \rho_a\,c_p VPD}{\gamma\,\lambda E_a} - \frac{(s + \gamma)\,r_{aH}}{\gamma} \tag{14}$$

We converted the $r_c$ estimates to canopy conductance $g_c$ (mm s$^{-1}$) using $g_c = 1000\ r_c^{-1}$ and will continue using $g_c$ (rather than $r_c$) for the remainder of the document. Note that the approach of calculating $kB^{-1}$ directly requires a separate
measurement of LW$_{in}$ and LW$_{out}$, which was only available in 95 of the 107 selected sites. For the remaining sites, an alternative approach was used to calculate $kB^{-1}$ (see Supporting Information).

### 2.4. Selection of unstressed days

To identify a subset of measurements per eddy-covariance site in which the ecosystem was unstressed we included two different approaches and provided the results for both methods. A first approach was based on soil moisture levels. For
those sites where soil moisture measurements were available, the maximal soil moisture level for each site was determined as the 98$^{th}$ percentile of all soil moisture measurements. We then split up the dataset of each site in 5 classes, according to the 20$^{th}$ percentiles of evaporation, in order to cover unstressed evaporation during all seasons. Days having soil moisture levels belonging to the highest 5% of soil moisture levels within each class were selected as unstressed days, but only if the soil moisture level of these selected days was above 75% of the maximum soil moisture.
As soil moisture data were not available for a large number of sites, and because using soil moisture data does not exclude days in which the functioning of the ecosystems has been affected by other kinds of biotic or abiotic stress, a second approach for defining unstressed days was applied based on an energy balance criterion. We calculated the EF from the daytime $\lambda E_a$ and H values, and considered it as a direct proxy for evaporative stress, i.e. we assumed that under unstressed conditions, a larger fraction of the available energy is used to evaporate (Gentine et al., 2007; 2011a; Maes et al., 2011).
This approach is similar to the one used in other $E_p$ studies on eddy-covariance or lysimeter data, in which the Bowen ratio (e.g. Douglas et al., 2009) or the ratio of $\lambda E_a/(SW_{in} + LW_{in})$ (Pereira and Pruitt, 2004) are used to define unstressed days. The unstressed record was comprised of all days with EF>95$^{th}$ percentile threshold for each particular site, or, if fewer than 15 days fulfilled this criterion, the 15 days with the highest EF. Consequently, we assume that at each site during at least 5% of the days the conditions are such that evaporation is unstressed and $E_a$ reflects $E_p$. The measured $E_a$
from the identified unstressed days is further referred to as $E_{unstr}$ (mm day$^{-1}$) and used as reference data to evaluate the different $E_p$ methods.

To assess whether the atmospheric conditions of the unstressed datasets are representative for the FLUXNET sites as such, a random bootstrap sample having the same number of records as the unstressed dataset was taken from the entire dataset of daily records and the mean, standard deviation, 2$^{nd}$ and 98$^{th}$ percentile of incoming shortwave radiation, air
temperature, vapour pressure deficit and wind speed was calculated. This procedure was repeated 1000 times. A t-test comparing the values of the unstressed subsample with those of the 1000 random samples was used to analyse whether the unstressed subsample was representative for the overall site conditions. This analysis was carried out for the soil moisture and for the energy balance criterion.




### 2.5. Calculation and calibration of the different $E_p$ methods

An overview of the different methods to calculate $E_p$ is given in Table 1. If possible, a reference crop, standard and biome-specific version of each method were calculated. The reference crop version calculates $E_p$ for the reference short turfgrass crop, and with the net radiation and other properties of this reference crop as well. The standard version uses the non-biome-specific parameters of the reference crop but considers the net radiation and other properties of the actual vegetation. The biome-specific version of each method applies a calibration of the key parameters (Table 1) and considers the radiation and other properties of the actual vegetation. These calibrated values by biome are based on the mean value of this key parameter for the unstressed dataset for each site, averaged out by biome type.

To estimate the radiation and crop properties of the reference crop versions, the equations described by Allen et al. (1998) in the FAO-56 method (Food and Agricultural Organization) were used and G was considered to be 0. $R_n$ was calculated as:

$$R_n=SW_{in} (1-\alpha_{ref}) + LW^* \tag{15}$$

with $\alpha_{ref}$=0.23 (Allen et al., 1998) and $LW^*$ the net longwave radiation, calculated after Allen et al. (1998; Eq. (39), Chapter 3).

In the case of the reference crop version of the Penman-Monteith equation (Eq. (1)), the FAO-56 method was used as described by Allen et al. (1998), with $g_{c\_ref}$ fixed as 14.49 mm s$^{-1}$ (corresponding with r$_{c\_ref}$ = 69 s m$^{-1}$) and using Eq. (15) to calculate R$_n$. The standard version of the Penman-Monteith equation used observed (R$_n$, G, VPD) and calculated ($s$, $\gamma$, $\rho_a$, r$_{aH}$) daytime values as described in Section 2.2. in Eq. (1), and assumed $g_{c\_ref}$ = 14.49 mm s$^{-1}$. The biome-specific version was calculated with the same data but used a biome-dependent value of $g_c$. First, for each individual site, the unstressed $g_c$ was calculated as the mean of the $g_c$ values of the unstressed record (see Section 2.4). The mean value per biome was then calculated from these unstressed $g_c$ values. Regarding the Penman method (Eq. (2)), the reference crop and standard versions were calculated using the same input data as for the Penman-Monteith methods; given Penman's consideration of no surface resistance, no biome-specific version was calculated.

The reference crop version of the Priestley and Taylor method is calculated from Eq. (3) with R$_n$ from Eq. 15, $s$ and $\gamma$ from the FAO-56 calculations, and with $\alpha_{PT}$ = 1.26. The standard version uses the same value for $\alpha_{PT}$ but the observed daytime values for R$_n$ and G. The biome-specific version followed a calibration of $\alpha_{PT}$ similar to the $g_{c\_ref}$ calculation. For each site, the unstressed $\alpha_{PT}$ was calculated as the average $\alpha_{PT}$, obtained by solving Eq. (3) for $\alpha_{PT}$ using the unstressed dataset. Finally, the mean per biome was calculated and used in the $E_p$ estimation. Regarding the method by Milly and Dunne (2017) (Eq. (4)), the reference crop, standard and biome-specific calculation were calculated accordingly, with R$_n$ from Eq. (15) for the reference crop version, $\alpha_{MD}$=0.8 for the reference crop and standard version, and a calibrated $\alpha_{MD}$ by biome type for the biome-specific version.

For Thornthwaite's and Oudin's formulations (Eqs. (5) and (6)), only the standard and biome-specific versions were calculated. The standard version of Thornthwaite's formulation uses $\alpha_{Th}$ = 16. In the biome-specific version, this parameter was again calculated per site as the mean value of the unstressed records (e.g. Xu and Singh, 2001;Bautista et al., 2009) and then averaged per biome. The effective temperature $T_{eff}$ was calculated as $T_{eff} = 0.36 (3T_{max} - T_{min})$ (Camargo et al., 1999). The parameter $b$ was calculated as $b = (6.75 \ 10^{-7}I^3)- (7.71 \ 10^{-7}I^2) + 0.0179 \ I + 0.492$ and the parameters $c$, $d$ and $e$ in Eq. (5c) were 415.85, 32.24 and 0.43, respectively (Pereira and Pruitt, 2004). Finally, for Oudin's temperature-based method, $\alpha_{Ou}$ = 100 was taken for the standard version (Eq. (6)). In the biome-specific version,



this value was recalculated by calculating $\alpha_{Ou}$ for the unstressed records through Eq. (6), calculating the mean $\alpha_{Ou}$ per site and finally the biome-dependent $\alpha_{Ou}$.

Altogether, this exercise yielded a total of 15 different methods to estimate $E_p$ whose specificities are documented in Table 1.

*(Insert Table 1)*

As mentioned, the biome-specific $E_p$ methods are calculated from the biome-specific means of $g_{c\_ref}$, $\alpha_{PT}$, $\alpha_{MD}$, $\alpha_{Th}$ and $\alpha_{Ou}$. In addition, the influence of climatic forcing data on $E_{unstr}$ and on these parameters was investigated. This was done

by calculating for each individual site the correlations between the daily estimates of climatic forcing data and the daily values of the unstressed datasets. Analyses were then performed on these correlations of all sites.

## 3. Results

### 3.1. Representativeness of climate forcing data of unstressed datasets

Table S2 provides an overview of the analyses used to verify if the climatic forcing data of the unstressed subsets were

representative for the forcing data of the sites as such. For the subsets of both unstressed criteria, climate forcing data were very representative for the site conditions, including for the VPD. For the energy balance criterion, for instance, only at one site, the unstressed subset of the 98th percentile was significantly different from the random sampling-based simulations and in only 2 sites, the 2nd percentile was significantly different from the simulations.

### 3.2. Key parameters by biome

We first focus on the parameter estimates of the unstressed record based on the energy balance criterion (Section 2.4). Of the full dataset, 107 flux sites meet all the selection criteria (i.e. at least 80 days without rainfall, good quality measurements of radiation and main fluxes, and at least 160 vegetation height observations – see Sections 2.2 and 2.3). Despite considerable variation within each biome, statistically significant differences are observed among biomes for all

of the key parameters of the unstressed records (see Sect. 2.3). Overall, croplands (CRO) are characterised by a higher measured $E_{unstr}$, which translates in the highest $g_{c\_ref}$, $\alpha_{PT}$, $\alpha_{MD}$, $\alpha_{Th}$ and the lowest $\alpha_{Ou}$ of all biomes. Deciduous broadleaf forest (DBF) and evergreen broadleaf forest (EBF) also have high $g_{c\_ref}$, $\alpha_{PT}$, $\alpha_{MD}$, $\alpha_{Th}$ and low $\alpha_{Ou}$, while savannah ecosystems (woody savannah (WSA), savannah (SAV) and open shrublands (OSH)) are characterised by lower $E_{unstr}$, and lower $g_{c\_ref}$, $\alpha_{PT}$, $\alpha_{MD}$, $\alpha_{Th}$ and higher $\alpha_{Ou}$. Only five sites (DE-KLI and IT-BCi, croplands; CA-SF3, OSH; AU-Rig,

grassland and AU-Wac, evergreen broadleaf forest) have mean values of $\alpha_{PT}$ higher than the typically assumed 1.26 (Table 2). In contrast, 27 sites, including 9 croplands, have a mean value of $\alpha_{MD}$ above 0.80, and 42 sites have mean $g_{c\_ref}$ above 14.49 mm s$^{-1}$. Finally, wetlands (WET) show a large standard deviation of $\alpha_{PT}$ and $\alpha_{MD}$ (Table 2) due to their location in tropical, temperate as well as in arctic regions.

Next, the effect of the climate forcing variables on $E_{unstr}$ and on the key parameters $g_{c\_ref}$, $\alpha_{PT}$ and $\alpha_{MD}$ of the unstressed

dataset is investigated. Fig. 3 gives the distribution of the correlations between the climate forcing variables and $E_{unstr}$, $g_{c\_ref}$, $\alpha_{PT}$ and $\alpha_{MD}$ of the unstressed records at each site. We did not include $\alpha_{Th}$ or $\alpha_{Ou}$ because the temperature-based methods did not perform well (see Section 3.3). $E_{unstr}$ was strongly positively correlated with $R_n$, $T_a$ and VPD in most





sites, but less with $u$ (Fig. 3a, Table 3). Considering all sites, the correlation between $g_{c\_ref}$ and the forcing variables is not significantly different from zero for any climate variable, yet $g_{c\_ref}$ is significantly negatively correlated with $T_{air}$ and with VPD in 40% and 45% of the flux tower sites, respectively (Table 3, Fig. 3b). The two other parameters, $\alpha_{PT}$ and $\alpha_{MD}$, appear less correlated to any of the climate variables across all sites (Table 3b). In the case of $\alpha_{MD}$, in particular, the

distributions of the correlations against all climate forcing variables peak around zero (Fig. 3d): $\alpha_{MD}$ is hardly influenced by $R_n$, and is overall not dependent on $u$, $T_a$, $[CO_2]$, or VPD in most sites (Table 3).

*(Insert Table 2)*

*(Insert Table 3)*

*(Insert Figure 3)*

### 3.3. Evaluation of different $E_p$ methods

We first list the results of the analysis using the energy balance criterion for selecting the unstressed records (Section 2.4). The scatterplots of measured $E_{unstr}$ versus estimated $E_p$ based on the 15 different methods are shown in Fig. 4 for three

sites belonging to different biomes. Despite the overall skill shown by the different $E_p$ methods, considerable differences can be appreciated. In general, the methods designed for reference crops (PM$_r$, Pe$_r$, PT$_r$, MD$_r$) overestimate $E_{unstr}$ and only two methods, MD$_B$ and PT$_B$, match the measured $E_{unstr}$ closely.

*(Insert Figure 4)*

Table 4 gives the mean correlation per biome for each method. The results are very consistent and reveal that the highest correlations for nearly all biomes are obtained with the standard and biome-specific radiation-based method (MD$_s$ and MD$_b$), closely followed by the standard and biome-dependent Priestley and Taylor method (PT$_s$ and PT$_b$). Temperature-based methods have the lowest overall mean correlation as well as lower mean correlations per biome. Note that the

correlations are the same for the standard and biome-specific version in the case of Priestley and Taylor (PT$_s$ and PT$_b$), radiation-based (MD$_s$ and MD$_b$) and Oudin (Ou$_s$ and Ou$_b$) methods (Table 4) – this is to be expected, as the only difference between the standard and biome-specific version of these methods is the value of their key parameters ($\alpha_{PT}$, $\alpha_{MD}$, $\alpha_{Ou}$) which are multiplicative (see Eqs. 3, 4 and 6). Differences are however reflected in their unbiased Root Mean Square Error (unRMSE) and mean bias – see Tables 5 and 6. The biome-specific versions of the radiation-based method (MD$_b$)

and of the Priestley and Taylor method (PT$_b$) have consistently the lowest unRMSE for all biomes. Though the difference between these two methods is small, MD$_b$ is performing slightly better. The standard Penman method (Pe$_s$) has the highest unRMSE. All reference crop methods (PM$_r$, Pe$_r$, PT$_r$, MD$_r$) have mean unRMSE above one, and the temperature-based methods (Th$_s$, Ou$_s$, Th$_b$, Ou$_b$) also have relatively high unRMSE. Finally, bias estimates are given in Table 6. Again, MD$_b$ is overall the best performing method (mean bias closest to 0), closely followed by the PT$_b$ method. Both methods have

consistently the lowest bias among all biomes, except for wetlands. Most reference crop methods (PM$_r$, Pe$_r$, PT$_r$, MD$_r$) as well as Pe$_s$ overestimate $E_p$ in all biomes.

*(Insert Table 4)*

*(Insert Table 5)*

*(Insert Table 6)*





The use of soil moisture content as criterion to select unstressed days (see Sect. 2.4) is explored. In total, 62 sites have soil water content data and meet the other selection criteria documented in Section 2.2. The results of this analysis are given in Tables S3–S5 of the supporting section. To allow for a fair comparison, the same statistics have also been

computed for just the same 62 tower sites with the energy balance-criterion (Tables S6-S8). Using the soil moisture criterion, the correlations are overall lower and the results of the mean correlation, unRMSE and biases are less consistent. However, the overall performance ranking of the different models remains similar: $PT_b$ is the best performing method with overall the highest mean correlation (R=0.84) and the lowest unRMSE (0.78 mm/day) and a bias closest to zero (-0.04), closely followed by the $MD_b$ method, with R=0.81, unRMSE=0.89 and a mean bias of -0.12. More complex

Penman-based models, and especially the empirical temperature-based formulations, show again a lower performance.

So far, all flux sites were used to calibrate the key parameters (Table 2) and those same sites were also used for the evaluation of the different methods. This was done to maximise the sample size. However, to avoid possible overfitting, we also repeated the analysis after separating calibration and validation samples. For each biome, two-thirds of the sites were randomly selected as calibration sites, and one third as validation sites. The key parameters were then calculated

from the calibration subset, and applied to estimate $E_p$ of the biome-specific methods of the validation subset. This procedure was repeated 100 times and the mean correlation, unbiased RMSE and bias per biome were calculated. Results show no substantial differences in overall correlation, unRMSE and bias of each method, and are provided in Tables S9–S11 for completeness.

## 4 Discussion

### 4.1. Comparison of criteria to define unstressed days

We prioritised the energy balance over the soil moisture criterion to select unstressed days, because it can be applied to sites without soil moisture measurements and because it implicitly allows the exclusion of days in which the ecosystem is stressed for reasons other than soil moisture availability (e.g. insect plagues, phenological leaf-out, fires, heat and atmospheric dryness stress, nutrient limitations). In addition, surface soil moisture at specific depths can be a poor

indicator of water stress, as rooting depth can vary and is not accurately measured, and different plants may exhibit various strategies and responses to drought (Powell et al., 2006;Douglas et al., 2009;Martínez-Vilalta et al., 2014). This is confirmed by our results: sampling unstressed days based on the energy balance-based criterion resulted in higher correlations (Table S6 vs Table S3) between $E_p$ and $E_{unstr}$ for all methods and in lower unRMSE, with the exception of the temperature-based methods (Table S7 vs Table S4). Therefore, in the following discussions the primary focus is on

the results of the energy balance method.

Nonetheless, as the MD method assumes a constant evaporative fraction, the use of the evaporative fraction as a criterion for selecting unstressed days can favour the MD as well as the closely related PT methods. Therefore, the soil moisture criterion taken here provides an independent check of the results, and confirms the robust and superior performance of the energy-driven $PT_b$ and $MD_b$ methods, which could otherwise be argued to have been favoured when selecting

unstressed days based on the energy balance.

### 4.2. Estimation of key ecosystem parameters

The biome-specific values of the key parameters in Table 2 are within the range of values used in reference crop and standard application of the models (Table 1), with the exception of $\alpha_{PT}$, which is typically lower than the standard value of 1.26. Other studies also found $\alpha_{PT}$ values far below 1.26 but within the range of our study, mainly for forests (e.g.





Shuttleworth and Calder, 1979;Viswanadham et al., 1991;Eaton et al., 2001;Komatsu, 2005), but also for tundra (Eaton et al., 2001) or grassland sites (Katerji and Rana, 2011) – see McMahon et al. (2013) for an overview. Our results and these studies demonstrate that the standard level of $\alpha_{PT}$=1.26 is close to the upper bound and will lead to an overestimation of $E_p$ at most sites (Table 5).

### 4.3. Performance of the Penman-Monteith method

The poor performance of the $PM_r$, $PM_s$ and $PM_b$ methods was relatively unexpected. Because the Penman-Monteith method incorporates the effects of air temperature, humidity, radiation and wind, it is often considered superior (e.g. Sheffield et al., 2012), and is even used as reference to evaluate other formulations (e.g. Xu and Singh, 2002;Oudin et al., 2005b;Sentelhas et al., 2010). However, in studies estimating $E_a$ at eddy-covariance tower sites, in which $g_c$ is adjusted so it reflects the actual stress conditions in the ecosystem, the Penman-Monteith method has already been shown to perform worse than other, simpler methods, such as the Priestley and Taylor equation (e.g. Ershadi et al., 2014;Michel et al., 2016). Its performance depends on the reliability of a wide range of input data, and on the methods used to derive $r_{aH}$ and $g_c$ (Singh and Xu, 1997;Dolman et al., 2014;Seiller and Anctil, 2016). In our case, the strict procedure followed to select the samples of 107 eddy-covariance datasets (see Sect. 2.2) ensured that all relevant variables were available, and that these meteorological measurements could be considered of high quality. Hence, in our analysis, poor input quality is unlikely to be the cause of low performance.

We believe that the underlying assumption of a constant $g_c$ typically adopted by PM methods ($PM_r$, $PM_s$, $PM_b$) when estimating $E_p$, is a more likely explanation of the poor performance. PM was the only method of which the biome-specific calibration did not improve its performance. This is partially because of the large variation in $g_{c\_ref}$ -values between the different flux sites of the same biome type (Table 2). In addition, of all the key parameters, $g_{c\_ref}$ values had the largest mean relative standard deviation of the unstressed datasets of the individual sites (*results not shown*). Canopy conductance of the unstressed dataset was significantly negatively correlated with VPD in 45% of the sites (Fig. 3b, Table 3). The relationship between $g_c$ and VPD for two such sites is illustrated in Fig. 5. It is clear that the $g_c$ data of unstressed days (red dots) are among the highest $g_c$ values for a given VPD, illustrating the validity of the energy balance method. However, it is also clear that $g_c$ of these unstressed days decreases sharply with increasing VPD. As a consequence, their mean value, used to ultimately calculate $g_{c\_ref}$ per biome type, is highly influenced by the local VPD and is not necessarily a representative value for this ecosystem.

The dependence of $g_c$ on VPD, even when soil moisture is not limiting (e.g. Jones, 1992;Granier et al., 2000;Sumner and Jacobs, 2005;Novick et al., 2016), has been well studied and incorporated in most conventional stomatal or canopy conductance models (e.g. Jarvis, 1976;Ball et al., 1987;Leuning, 1995). Yet, out of practical reasons, $g_{c\_ref}$ is usually considered constant in $E_p$-methods using the Penman-Monteith approach, with the $PM_r$ as best illustration. Our data confirm that stomata close when VPD increases, even in unstressed conditions. As such, the VPD-dependence of $g_c$ smooths the impact of VPD in the Penman-Monteith equation: if VPD gets high, $g_c$ becomes very low, lowering the impact of VPD on $E_a$ (Eq. 1). Assuming a constant $g_{c\_ref}$ value overestimates the influence of VPD (and wind speed) on $E_p$.

Apart from the VPD-dependence, assuming a constant $g_{c\_ref}$-value in the Penman-Monteith method also ignores the effect of increasing $CO_2$ levels on $g_c$. As a result, Milly and Dunne (2016) found that the Penman-Monteith methods with constant $g_{c\_ref}$ overpredicted $E_p$ in models estimating future water use. Incorporating a VPD and $CO_2$ calibration of $g_{c\_ref}$ in the Penman-Monteith equation is outside the scope of this study, but can be an interesting approach to further improve $E_p$ calculations. However, it would make the model even more complex and requires more empirical input data. Likewise,



taking a wet canopy as reference in the Penman method ($g_c = \infty$ or $r_c=0$), not only severely overestimates $E_p$ (Table 6) but also overestimates the influence of VPD and wind speed on $E_p$.

*(Insert Figure 5)*

### 4.4. Considerations regarding simple energy-based methods

The simpler Priestley and Taylor and radiation-based methods came forward as the best methods for assessing $E_p$ with both criteria to define unstressed days. These observations are in agreement with studies highlighting radiation as the dominant driver of evaporation of saturated or unstressed ecosystems (e.g. Priestley and Taylor, 1972;Abtew,
1996;Wang et al., 2007;Song et al., 2017;Chan et al., 2018). It also agrees with Douglas et al. (2009) who found that PT outperformed the PM method for estimating unstressed evaporation in 18 FLUXNET sites.

Both $PT_b$ and $MD_b$ are attractive from a modelling perspective, as they require a minimum of input data. However, this simplicity can also hold risks. The Priestley and Taylor method has been criticised for the implicit assumptions, which are also present in the $MD_b$-method. For instance, by not incorporating wind speed explicitly, it is assumed that the effect
of wind speed on $E_p$ is somehow embedded within $\alpha_{PT}$ (or $\alpha_{MD}$). Yet, several studies indicate that wind speeds are decreasing ('stilling') globally (McVicar et al., 2008;Vautard et al., 2010;McVicar et al., 2012). McVicar et al. (2012) also reported an associated decreasing trend in observed pan evaporation worldwide as well as in $E_p$ calculated with the $PM_r$ method. As Priestley and Taylor methods do not incorporate effects of wind speed, they cannot capture these trends. A similar criticism can be drawn with regards to the effect of $[CO_2]$ on stomatal conductance, water use efficiency, and
thus potential transpiration (Field et al., 1995). A separate question is whether more complex $E_p$ methods that incorporate the effects of wind speed, $[CO_2]$ or VPD explicitly do this correctly; the above-mentioned issues about the fixed parameterisation of the Penman-Monteith methods for estimating $E_p$ indicate that this may typically not be the case. In addition, both Penman methods showed a relatively poor performance and severely overestimated $E_{unstr}$ (Tables 4 and 6; S3 and S5).

Regarding the non-explicit consideration of wind speed by simpler methods, our records show a limited effect of $u$ on $E_a$ and $E_p$, even when considering larger temporal scales. Of the 16 flux towers with at least 10 years of evaporation data, we calculated the yearly average $E_a$ as well as the annual mean climatic forcing variables. Yearly averages were calculated from monthly averages, which in turn were calculated if at least three daytime measurements were available. Despite a relatively large mean standard deviation in yearly average $u$ of 7.0%, yearly average $u$ was not significantly correlated
with $E_a$ in any of these sites. In contrast, yearly average net radiation was positively correlated with yearly average $E_a$ in seven of the 16 sites, with comparable mean standard deviation in annual $R_n$ (8.5%). Moreover, looking at all individual towers and using the daily estimates, neither $\alpha_{MD}$ nor $\alpha_{PT}$ were correlated with wind speed (Fig. 3c, d, Table 3). In fact, since $\alpha_{MD}$ appears hardly affected by any climatic variable, and given the relatively small range in $\alpha_{MD}$ values within each biome (Table 2), it appears that $\alpha_{MD}$ is a robust biome property that can be used in the seamless application of these
methods at global scale. The robustness of $\alpha_{MD}$ as biome property is furthermore confirmed by the analysis with independent calibration and validation sites, which hardly affected the unRMSE and bias (Tables S10–11).





### 4.5. Use of available energy under stress conditions

The best performing methods rely heavily on observations of the available energy ($R_n$–G) (Eqs. 2 and 3). In Section 3.2, all $E_p$ calculations used available energy obtained during unstressed conditions. The question is whether $E_p$ can also be calculated correctly using the observed radiation fluxes and G when the ecosystems are under stress. As mentioned in

Section 1, while several authors argued that the net radiation of the reference system may be used to calculate $E_p$ of the actual ecosystem, others considered only incoming shortwave and longwave radiation as forcing variables, while treating outgoing longwave and shortwave radiation as ecosystem responses (e.g. Lhomme, 1997;Lhomme and Guilioni, 2006;Shuttleworth, 1993). Among other considerations, it is clear that $T_s$ will be lower if vegetation is healthy and soils are well watered (Maes and Steppe, 2012), which results in lower outgoing longwave radiation and higher net radiation.

Therefore, while using the observations of incoming shortwave and longwave radiation in the computation of $E_p$ is defendable (despite the potential atmospheric feedbacks that may derive from the consideration of unstressed conditions), the outgoing fluxes and G should be recomputed for the expected α, $T_s$ and G during unstressed conditions. Note that when doing so, the $E_p$ method deviates slightly from $E_{p0}$ in the complementarity relationship, in which atmospheric feedbacks affecting incoming radiation or VPD are implicitly considered (Kahler and Brutsaert, 2006).

A method to derive the expected values of unstressed α, $T_s$ and G under stressed conditions is presented in the Section S2 of the Supporting Information and is based the $MD_b$ method and on flux tower data of the unstressed datasets. However, this method requires a large amount of input data and is not practically applicable at global scale. Comparing $E_p$ obtained with this method to $E_p$ calculated with the actual ($R_n$-G) reveals that the estimation of the latter method results in $E_p$ values that are 8.2±10.1% lower. There are no significant differences between biomes, but the distribution of the underestimation

is left-skewed: although the median underestimation is 5.5%, the underestimation is >10% in 22% of the sites (Fig. 6). The main reason, explaining about 65% of this underestimation, is the difference between the actual $T_s$ observations and the $T_s$ calculated for hypothetical unstressed condition. A straightforward alternative to match $E_p$ of the modelled data based on flux tower data with $E_p$ when only data of temperature and radiation are available is to assume that the unstressed $T_s$ can be approximated as the mean of $T_a$ and the actual $T_s$, in which case $E_p$ can be calculated with the $MD_b$ method as

$$E_p = \ \alpha_{MD} \left((1 - \alpha)\ SW_{in} + \ \varepsilon\ LW_{in} - 0.5\ \varepsilon\ LW_{out} - \ 0.5\ \varepsilon\ \sigma\ T_a^4 - \ G\right) \tag{16}$$

This approach was also tested and resulted in a slight mean underestimation of $E_p$ of 2.6 ± 5.8% compared to $E_p$ calculated with the flux tower data and estimating unstressed α, $T_s$ and G, but with a mean median value at -0.1% (Fig. 6). Given the low error and the straightforward calculation, we recommend this method to calculate $E_p$ at global scales.

*(Insert Figure 6)*

Fig. 7 gives an example of the seasonal evolution of $E_a$ and $E_p$ and the $S$ factor ($S = E_a E_p^{-1}$) in a grassland (Fig 7a) and a deciduous forest site (Fig 7a). The short growing season in the grassland site, when $E_a$ is close to $E_p$ and values of $S$ are close to 1, stands in clear contrast with the winter period, when grasses have died off and $E_a$ and consequently also $S$ are very low. In the relatively wet broadleaf forest, $E_a$ and $E_p$ follow a similar seasonal cycle. In winter, when total evaporation

is limited to soil evaporation, $S$ is very low; in spring, when leaves are still developing, $E_a$ lags $E_p$. In summer, $S$ remain high and close to one.

*(Insert Figure 7)*





## Conclusion

Based on a large sample of eddy-covariance sites from the FLUXNET2015 database, we demonstrated a higher potential of radiation-driven methods calibrated by biome type to estimate $E_p$ than of more complex Penman-Monteith approaches or empirical temperature-based formulations. This was consistent across all 11 biomes represented in the database, and

for two different criteria to identify unstressed days, one based on soil moisture and the other on evaporative fractions. Our analyses also showed that the key parameters required to apply the higher-performance radiation-driven methods are relatively insensitive to climate forcing. This makes these methods robust for incorporation into global offline models, e.g. for hydrological applications. Finally, we conclude that, at the ecosystem scale, Penman-Monteith methods for estimating $E_p$ should only be prioritised if the unstressed stomatal conductance is calculated dynamically (e.g. as a

function of LAI, VPD) and high accuracy observations from the wide palate of required forcing variables are available.

## Data availability

The FLUXNET2015 dataset can be downloaded from http://fluxnet.fluxdata.org/data/fluxnet2015-dataset/. The main script for calculating potential evaporation with the different method as well as the daily flux data of one site (AU-How), for which permission of distribution was granted, is available as supplement. For further questions, we ask readers to

contact the corresponding author at wh.maes@ugent.be.

## Acknowledgements

This study was funded by the Belgian Science Policy Office (BELSPO) in the frame of the STEREO III program project STR3S (SR/02/329). Diego G. Miralles acknowledges support from the European Research Council (ERC) under grant agreement no. 715254 (DRY–2–DRY). This work used eddy covariance data acquired and shared by the FLUXNET

community, including these networks: AmeriFlux, AfriFlux, AsiaFlux, CarboAfrica, CarboEuropeIP, CarboItaly, CarboMont, ChinaFlux, FLUXNET-Canada, GreenGrass, ICOS, KoFlux, LBA, NECC, OzFlux-TERN, TCOS-Siberia, and USCCC. The ERA-Interim reanalysis data are provided by ECMWF and processed by LSCE. The FLUXNET eddy covariance data processing and harmonization was carried out by the European Fluxes Database Cluster, AmeriFlux Management Project, and Fluxdata project of FLUXNET, with the support of CDIAC and ICOS Ecosystem Thematic

Center, and the OzFlux, ChinaFlux and AsiaFlux offices. The Atqasuk and Ivotuk towers in Alaska were supported by the Office of Polar Programs of the National Science Foundation (NSF) awarded to DZ, WCO, and DAL (award number 1204263) with additional logistical support funded by the NSF Office of Polar Programs, and by the Carbon in Arctic Reservoirs Vulnerability Experiment (CARVE), an Earth Ventures (EV-1) investigation, under contract with the National Aeronautics and Space Administration, and by the ABoVE (NNX15AT74A; NNX16AF94A) Program. The OzFlux

network is supported by the Australian Terrestrial Ecosystem Research Network (TERN, http://www.tern.org.au). We would also like to thank three anonymous reviewers for their useful suggestions on an earlier version of the manuscript.

**Author contribution.** WHM and DGM designed the research. PG and NECV assisted in developing the optimal method for analysing all flux tower data. WHM performed the calculations and analyses. WHM and DGM prepared the

manuscript, with contributions from PG and NECV.





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





**Tables**

Abbreviation list

| Symbol | Description | Unit |
|---|---|---|
| $\alpha$ | Albedo | [-] |
| $\alpha_{MD}$ | Parameter of Milly & Dunne equation (Eq. (4)) | [-] |
| $\alpha_{Ou}$ | Parameter of Oudin equation (Eq. (6)) | [-] |
| $\alpha_{PT}$ | Parameter of Priestley & Taylor equation (Eq. (3)) | [-] |
| $\alpha_{ref}$ | Albedo of reference crop (0.23) | [-] |
| $\alpha_{Th}$ | Parameter of Thornthwaite method (Eq. (5b)) | [-] |
| $c_p$ | Specific heat capacity of the air | J kg$^{-1}$ K$^{-1}$ |
| $d$ | Zero displacement height | m |
| $\varepsilon$ | Emissivity | [-] |
| $E$ | Ecosystem evaporation, (or evapotranspiration, the sum of soil evaporation, transpiration, interception evaporation and snow sublimation) | kg m$^{-2}$ s$^{-1}$ or mm day$^{-1}$ |
| $E_a$ | Actual evaporation | kg m$^{-2}$ s$^{-1}$ or mm day$^{-1}$ |
| $E_p$ | Potential evaporation | kg m$^{-2}$ s$^{-1}$ or mm day$^{-1}$ |
| $E_{p0}$ | Evaporation from an extensive well-watered surface (Complementary Relationship) | mm day$^{-1}$ |
| $E_{pa}$ | Evaporation from an infinitesimally small well-watered surface (Complementary Relationship) | mm day$^{-1}$ |
| $E_{unstr}$ | Evaporation from an unstressed ecosystem | mm day$^{-1}$ |
| EF | Evaporative fraction | [-] |
| $\gamma$ | Psychrometric constant | Pa K$^{-1}$ |
| G | Ground heat flux | W m$^{-2}$ |
| $g$ | Gravitational acceleration | m s$^{-2}$ |
| $g_c$ | Canopy conductance to water transfer | m s$^{-1}$ or mm s$^{-1}$ |
| $g_{c\_ref}$ | $g_c$ of reference crop | mm s$^{-2}$ |
| H | Sensible heat flux | W m$^{-2}$ |
| I | parameter in Thornthwaite equation (Eq. (5)) | |
| $k$ | von Karman constant (0.41) | [-] |
| $kB^{-1}$ | Stanton number | [-] |
| $\lambda$ | Latent heat of vaporisation | J kg$^{-1}$ |
| L | Obukhov length | m |
| $LW_{in}$ | Incoming longwave radiation | W m$^{-2}$ |
| $LW_{out}$ | Outgoing longwave radiation | W m$^{-2}$ |
| $MD_b$ | Biome-specific version of the Milly-Dunne method (Table 1) | |
| $MD_r$ | Reference crop version of the Milly-Dunne method (Table 1) | |
| $MD_s$ | Standard version of the Milly-Dunne method (Table 1) | |
| $Ou_b$ | Biome-specific version of the Oudin method (Table 1) | |
| $Ou_s$ | Standard version of the Oudin method (Table 1) | |
| $Pe_r$ | Reference crop version of the Penman method (Table 1) | |
| $Pe_s$ | Standard version of the Penman method (Table 1) | |
| $PM_b$ | Biome-specific version of the Penman-Monteith method (Table 1) | |





| | | |
|---|---|---|
| $PM_r$ | Reference crop version of the Penman-Monteith method (Table 1) | |
| $PM_s$ | Standard version of the Penman-Monteith method (Table 1) | |
| $PT_b$ | Biome-specific version of the Priestley and Taylor method (Table 1) | |
| $PT_r$ | Reference crop version of the Priestley and Taylor method (Table 1) | |
| $PT_s$ | Standard version of the Priestley and Taylor method (Table 1) | |
| $q_a$ | Specific humidity | kg kg$^{-1}$ |
| $\rho_a$ | Air density | kg m$^{-3}$ |
| $\Psi_h(X)$ | Businger-Dyer stability function for heat exchange of variable X | [-] |
| $\Psi_m(X)$ | Businger-Dyer stability function for momentum of variable X | [-] |
| $r_{aH}$ | Resistance of heat transfer to air | s m$^{-1}$ |
| $r_c$ | Canopy resistance of water transfer | s m$^{-1}$ |
| $R_e$ | Top-of-atmosphere radiation | MJ m$^{-2}$ day$^{-1}$ |
| Re | Reynolds number | [-] |
| RH | Relative humidity | % |
| $RH_{min}$ | Minimum daily RH | % |
| $RH_{max}$ | Maximum daily RH | % |
| $R_n$ | Net radiation | W m$^{-2}$ |
| $\sigma$ | Stefan-Boltzmann constant (5.67 10$^{-8}$) | W m$^{-2}$ K$^{-4}$ |
| $s$ | slope of the Clausius-Clapeyron curve | Pa K$^{-1}$ |
| $S$ | Ratio of $E_a$ and $E_p$, confined to [0-1] | [-] |
| $SW_{in}$ | Incoming shortwave radiation | W m$^{-2}$ |
| $SW_{out}$ | Outgoing shortwave radiation | W m$^{-2}$ |
| $SW_{TOA}$ | $SW_{in}$ at the top of atmosphere | W m$^{-2}$ |
| $T_0$ | Aerodynamic temperature | °C or K |
| $T_a$ | Air temperature | °C |
| $T_{a\_mean}$ | Mean air temperature for each month | °C |
| $T_{eff}$ | Effective temperature (Thornthwaite equation, Eq. (5)) | °C |
| $Th_b$ | Biome-specific version of the Thornthwaite method (Table 1) | |
| $Th_s$ | Standard version of the Thornthwaite method (Table 1) | |
| $T_{min}$ | Minimum daily $T_a$ | °C |
| $T_{max}$ | Maximum daily $T_a$ | °C |
| $T_s$ | Surface temperature | °C or K |
| VH | Vegetation height | m |
| VPD | Vapour pressure deficit | Pa or hPa |
| $u$ | Wind speed | m s$^{-1}$ |
| $u*$ | Friction velocity | m s$^{-1}$ |
| $z$ | Wind sensor height | m |
| $z_{0h}$ | Roughness length for heat exchange | m |
| $z_{0m}$ | Roughness length for momentum | m |





**Table 1. Overview of the different $E_p$ methods used in this study and their calculation.**

| | | Key parameter | $R_n$ | | $r_{aH}$ | $T_a$ | RH or VPD | N |
|---|---|---|---|---|---|---|---|---|
| | | | SW* | LW* | | | | |
| **Penman-Monteith** | | $g_{c\_ref}$ (mm s⁻¹) | | | | | | |
| PM$_r$ | Reference crop | 14.49 | FAO-56 ($\alpha_{ref}$=0.23) | FAO-56 $f$(T$_{max}$,T$_{min}$, SW, SW$_{TOA}$, e$_a$) | 208 u$_{2m}$⁻¹ | From T$_{max}$,T$_{min}$ | From RH$_{Max}$, RH$_{Min}$ | FAO-56 |
| PM$_s$ | Standard | 14.49 | measured | measured | calculated | Daytime mean | Daytime mean | |
| PM$_b$ | Biome-specific | Biome-specific | measured | measured | calculated | Daytime mean | Daytime mean | |
| **Penman** | | $g_{c\_ref}$ (mm s⁻¹) | | | | | | |
| Pe$_r$ | Reference crop | ∞ (r$_{c\_ref}$ = 0) | FAO-56 ($\alpha_{ref}$ =0.23) | FAO-56 $f$(T$_{max}$,T$_{min}$, SW, SW$_{TOA}$, e$_a$) | 208 u$_{2m}$⁻¹ | From T$_{max}$,T$_{min}$ | From RH$_{Max}$, RH$_{Min}$ | |
| Pe$_s$ | Standard | ∞ (r$_{c\_ref}$ = 0) | measured | measured | calculated | Daytime mean | Daytime mean | |
| **Priestley and Taylor** | | $\alpha_{PT}$ (-) | | | | | | |
| PT$_r$ | Reference crop | 1.26 | FAO-56 ($\alpha_{ref}$ =0.23) | FAO-56 $f$(T$_{max}$,T$_{min}$, SW, SW$_{TOA}$, e$_a$) | | | | |
| PT$_s$ | Standard | 1.26 | measured | measured | | Daytime mean | | |
| PT$_b$ | Biome-specific | Biome-specific | measured | measured | | Daytime mean | | |
| **Milly and Dunne** | | $\alpha_{MD}$ (-) | | | | | | |
| MD$_r$ | Reference crop | 0.8 | FAO-56 ($\alpha_{ref}$ =0.23) | FAO-56 $f$(T$_{max}$,T$_{min}$, SW, SW$_{TOA}$, e$_a$) | | | | |
| MD$_s$ | Standard | 0.8 | measured | measured | | | | |
| MD$_b$ | Biome-specific | Biome-specific | measured | measured | | | | |
| **Thornthwaite** | | $\alpha_{Th}$ (-) | | | | | | |
| Th$_s$ | Standard | 16 | | | | From T$_{max}$,T$_{min}$ | | Measured |
| Th$_b$ | Biome-specific | Biome-specific | | | | From T$_{max}$,T$_{min}$ | | Measured |
| **Oudin** | | $\alpha_{ou}$ | | | | | | |
| Ou$_s$ | Standard | 100 | | | | Daily mean | | |
| Ou$_b$ | Biome-specific | Biome-specific | | | | Daily mean | | |

N=number of daylight hours; T$_{max}$, T$_{min}$, RH$_{max}$ and RH$_{min}$ the maximum and minimum daily air temperature or RH; SW* and LW* are net shortwave and net longwave radiation; SW$_{TOA}$ is the shortwave incoming radiation at the top the atmosphere; FAO-56 refers to the methodology described by Allen et al. (1998).





Table 2. Overview of the difference of the key parameters ($g_{c\_ref}$, $\alpha_{PT}$, $\alpha_{MD}$, $\alpha_{Th}$ and $\alpha_{Ou}$) during unstressed conditions per biome. The energy balance method was used for defining unstressed days (See section 2.4, see Table 1 for definition of key parameters). The *p* value of the ANOVA test is given, as well as the mean ± 1 standard deviation for each biome. Different alphabetic superscripts indicate significantly differing means (Tukey post-hoc test; $p<0.05$). The number of sites per biome is given between brackets. Different colours are used to group biomes into broader ecosystem types (in descending order: croplands, grasslands, forests, savannah ecosystems, wetlands).

| | $g_{c\_ref}$ (mm s$^{-1}$) | $\alpha_{PT}$ (-) | $\alpha_{RB}$ (-) | $\alpha_{Th}$ (-) | $\alpha_{Ou}$ (-) |
|---|---|---|---|---|---|
| *p* (ANOVA) | 0.017 | 0.004 | <0.001 | <0.001 | <0.001 |
| CRO (10) | 38.3 ± 23.0 | 1.15 ± 0.14[a] | 0.86 ± 0.09[a] | 38.7 ± 14.5[ab] | 77.0 ± 27.8[b] |
| GRA (20) | 30.5 ± 40.2 | 1.02 ± 0.16[ab] | 0.74 ± 0.12[ab] | 30.4 ± 13.9[ab] | 103.2 ± 38.9[ab] |
| DBF (15) | 32.6 ± 27.4 | 1.09 ± 0.14[ab] | 0.80 ± 0.08[ab] | 33.3 ± 7.8[ab] | 70.5 ± 18.0[ab] |
| EBF (9) | 42.0 ± 36.6 | 1.09 ± 0.15[ab] | 0.74 ± 0.05[a] | 53.1 ± 16.8[a] | 95.5 ± 22.9[ab] |
| ENF (26) | 28.4 ± 52.1 | 0.89 ± 0.26[ab] | 0.62 ± 0.09[ab] | 40.3 ± 16.7[ab] | 92.0 ± 21.8[ab] |
| MF (4) | 10.0 ± 7.1 | 0.88 ± 0.23[b] | 0.64 ± 0.13[b] | 26.1 ± 3.6[ab] | 138.2 ± 91.6[ab] |
| CSH (2) | 8.5 ± 3.9 | 0.90 ± 0.10[ab] | 0.64 ± 0.15[ab] | 41.4 ± 13.7[ab] | 130.3 ± 36.1[a] |
| WSA (5) | 8.4 ± 3.4 | 0.95 ± 0.09[ab] | 0.70 ± 0.10[ab] | 33.8 ± 6.4[ab] | 104.6 ± 19.7[ab] |
| OSH (5) | 7.8 ± 3.7 | 0.87 ± 0.14[ab] | 0.68 ± 0.15[b] | 35.0 ± 4.1[ab] | 147.1 ± 63.9[ab] |
| SAV (6) | 4.3 ± 2.0 | 0.79 ± 0.11[b] | 0.58 ± 0.09[ab] | 31.3 ± 11.2[ab] | 147.7 ± 61.8[ab] |
| WET (5) | 20.0 ± 14.1 | 1.03 ± 0.47[ab] | 0.75 ± 0.11[ab] | 17.8 ± 13.3[b] | 638.6 ± 1230.1[ab] |

CRO=cropland; DBF=Deciduous Broadleaf Forest; EBF=Evergreen Broadleaf Forest; ENF=Evergreen Needleleaf Forest; MF=Mixed Forest; CSH=Closed Shrubland; WSA=Woody Savanna; SAV=Savanna; OSH=Open Shrubland; GRA=Grasslands; WET=Wetlands.





**Table 3. Influence of climate forcing variables on $E_{unstr}$ and selected key parameters ($g_{c\_ref}$, $\alpha_{PT}$, $\alpha_{MD}$). (left) Mean ± 1 standard deviation of the correlations of $E_{unstr}$, $g_c$, $\alpha_{PT}$ and $\alpha_{MD}$ against the climate forcing variables, and (right) number of sites (out of total = 107) with significant negative/positive correlations between $E_{unstr}$, $\alpha_{PT}$, $g_{c\_ref}$ and $\alpha_{MD}$ and the climate forcing variables. Based on unstressed days only defined using the energy balance criterion.**

| | Mean ± 1 standard deviation of the correlations | | | | Number of sites with significant negative/positive correlations | | | |
|---|---|---|---|---|---|---|---|---|
| | $E_{unstr}$ | $g_{c\_ref}$ | $\alpha_{PT}$ | $\alpha_{MD}$ | $E_{unstr}$ | $g_{c\_ref}$ | $\alpha_{PT}$ | $\alpha_{MD}$ |
| Wind | $0.13 \pm 0.26$ | $0.03 \pm 0.25$ | $0.12 \pm 0.31$ | $0.01 \pm 0.22$ | 6/26 | 4/13 | 11/30 | 5/6 |
| $T_{air}$ | $0.60 \pm 0.24^{*}$ | $-0.22 \pm 0.29$ | $-0.21 \pm 0.34$ | $-0.02 \pm 0.28$ | 0/93 | 43/0 | 43/5 | 16/13 |
| VPD | $0.64 \pm 0.20^{*}$ | $-0.27 \pm 0.27$ | $-0.11 \pm 0.31$ | $-0.01 \pm 0.28$ | 0/93 | 48/0 | 31/10 | 15/11 |
| $R_n$ | $0.90 \pm 0.08^{*}$ | $-0.05 \pm 0.25$ | $-0.13 \pm 0.30$ | $-0.10 \pm 0.31$ | 0/106 | 17/3 | 33/5 | 30/14 |
| $[CO_2]$ | $-0.16 \pm 0.30$ | $-0.01 \pm 0.23$ | $-0.03 \pm 0.22$ | $-0.03 \pm 0.25$ | 34/5 | 7/5 | 9/4 | 12/4 |

5   *significantly different from 0




**Table 4. Mean correlations per biome between the measured $E_{\text{unstr}}$ and the different $E_p$ methods. The methods with the highest correlation per biome are highlighted in bold and underlined. Based on unstressed days only defined using the energy balance criterion. Different colours are used to group biomes into broader ecosystem types (in descending order: croplands, grasslands, forests, savannah ecosystems, wetlands).**

| | Radiation, Temperature, VPD | | | | | Radiation, Temperature | | | Radiation | | | Temperature | | | |
| | Penman-Monteith | | | Penman | | Priestley and Taylor | | | Milly and Dunne | | | Thornthwaite | | Oudin | |
| | Ref. crop | Standard | Biome | Ref. crop | Standard | Ref. crop | Standard | Biome | Ref. crop | Standard | Biome | Standard | Biome | Standard | Biome |
|---|---|---|---|---|---|---|---|---|---|---|---|---|---|---|---|
| CRO (10) | 0.84 | 0.91 | 0.90 | 0.76 | 0.81 | 0.86 | 0.96 | 0.96 | 0.82 | **0.96** | **0.96** | 0.77 | 0.77 | 0.74 | 0.74 |
| GRA (20) | 0.79 | 0.87 | 0.87 | 0.77 | 0.84 | 0.82 | 0.93 | 0.93 | 0.80 | **0.94** | **0.94** | 0.55 | 0.54 | 0.55 | 0.55 |
| DBF (15) | 0.78 | 0.87 | 0.88 | 0.79 | 0.85 | 0.78 | 0.91 | 0.91 | 0.75 | **0.91** | **0.91** | 0.57 | 0.56 | 0.57 | 0.57 |
| EBF (9) | 0.88 | 0.89 | 0.88 | 0.86 | 0.85 | 0.87 | **0.91** | **0.91** | 0.83 | 0.90 | 0.90 | 0.71 | 0.79 | 0.57 | 0.57 |
| ENF (26) | 0.89 | 0.90 | 0.91 | 0.88 | 0.86 | 0.90 | 0.95 | 0.95 | 0.88 | **0.95** | **0.95** | 0.77 | 0.79 | 0.76 | 0.76 |
| MF (4) | 0.90 | 0.93 | 0.93 | 0.90 | 0.93 | 0.90 | **0.94** | **0.94** | 0.88 | 0.93 | 0.93 | 0.79 | 0.75 | 0.74 | 0.74 |
| CSH (2) | 0.90 | 0.94 | 0.93 | 0.89 | 0.90 | 0.90 | 0.95 | 0.95 | 0.89 | **0.95** | **0.95** | 0.80 | 0.78 | 0.75 | 0.75 |
| WSA (5) | 0.76 | 0.78 | 0.78 | 0.73 | 0.73 | 0.80 | 0.89 | 0.89 | 0.79 | **0.90** | **0.90** | 0.41 | 0.41 | 0.46 | 0.46 |
| SAV (6) | 0.79 | 0.82 | 0.81 | 0.77 | 0.79 | 0.83 | **0.91** | **0.91** | 0.81 | 0.91 | 0.91 | 0.52 | 0.52 | 0.56 | 0.56 |
| OSH (5) | 0.72 | 0.80 | 0.78 | 0.64 | 0.78 | 0.79 | 0.90 | 0.90 | 0.77 | **0.90** | **0.90** | 0.54 | 0.53 | 0.56 | 0.56 |
| WET (5) | 0.87 | 0.81 | 0.76 | 0.87 | 0.66 | 0.79 | 0.83 | 0.83 | 0.68 | **0.85** | **0.85** | 0.50 | 0.45 | 0.61 | 0.61 |
| Overall (107) | 0.83 | 0.87 | 0.87 | 0.81 | 0.83 | 0.84 | 0.92 | 0.92 | 0.81 | **0.93** | **0.93** | 0.62 | 0.63 | 0.63 | 0.63 |

CRO=cropland; DBF=Deciduous Broadleaf Forest; EBF=Evergreen Broadleaf Forest; ENF=Evergreen Needleleaf Forest; MF=Mixed Forest; CSH=Closed Shrubland; WSA=Woody Savanna; SAV=Savanna; OSH=Open Shrubland; GRA=Grasslands; WET=Wetlands.





**Table 5. Unbiased Root Mean Square Error (UnRMSE) (in mm day$^{-1}$) for the $E_p$ methods per biome. The methods with the lowest UnRMSE per biome are indicated in bold and are underlined. Based on unstressed days only defined using the energy balance criterion. Different colours are used to group biomes into broader ecosystem types (in descending order: croplands, grasslands, forests, savannah ecosystems, wetlands).**

| | Radiation, Temperature, VPD | | | | | Radiation, Temperature | | | Radiation | | | Temperature | | | |
| | Penman-Monteith | | | Penman | | Priestley and Taylor | | | Milly and Dunne | | | Thornthwaite | | Oudin | |
| | Ref. crop | Standard | Biome | Ref. crop | Penman | Ref. crop | Standard | Biome | Ref. crop | Standard | Biome | Standard | Biome | Standard | Biome |
|---|---|---|---|---|---|---|---|---|---|---|---|---|---|---|---|
| CRO (10) | 1.16 | 0.79 | 1.04 | 1.60 | 2.88 | 1.27 | 0.62 | 0.58 | 1.21 | 0.57 | **0.55** | 1.24 | 1.24 | 1.29 | 1.27 |
| GRA (20) | 1.22 | 0.70 | 0.81 | 1.75 | 1.04 | 1.40 | 0.58 | 0.47 | 1.13 | 0.44 | **0.44** | 1.07 | 1.03 | 1.05 | 1.04 |
| DBF (15) | 1.14 | 0.88 | 0.89 | 1.21 | 1.36 | 1.29 | 0.75 | **0.72** | 1.20 | 0.72 | 0.72 | 1.41 | 1.42 | 1.37 | 1.32 |
| EBF (9) | 0.84 | 0.62 | 0.93 | 1.07 | 1.33 | 1.09 | 0.75 | 0.59 | 0.96 | 0.55 | **0.54** | 1.04 | 0.98 | 1.15 | 1.14 |
| ENF (26) | 0.98 | 0.78 | 0.99 | 1.20 | 14.89 | 1.26 | 0.84 | 0.52 | 1.09 | 0.59 | **0.50** | 0.94 | 0.91 | 0.96 | 0.97 |
| MF (4) | 1.23 | 0.69 | 0.69 | 1.58 | 1.11 | 1.64 | 0.86 | **0.58** | 1.26 | 0.64 | 0.59 | 1.11 | 1.03 | 1.03 | 0.99 |
| CSH (2) | 0.82 | 0.59 | 0.59 | 0.98 | 0.92 | 1.12 | 0.75 | **0.48** | 0.91 | 0.55 | 0.49 | 0.90 | 0.96 | 0.83 | 0.81 |
| WSA (5) | 1.15 | 0.93 | 0.80 | 1.41 | 1.68 | 1.27 | 0.67 | 0.52 | 1.00 | 0.53 | **0.51** | 1.10 | 1.10 | 0.99 | 0.99 |
| SAV (6) | 1.22 | 1.02 | 0.83 | 1.53 | 1.88 | 1.39 | 0.76 | 0.52 | 1.07 | 0.58 | **0.52** | 1.22 | 1.21 | 1.10 | 0.97 |
| OSH (5) | 1.37 | 0.73 | 0.63 | 1.94 | 0.92 | 1.63 | 0.67 | **0.43** | 1.28 | 0.48 | 0.44 | 1.12 | 1.03 | 0.90 | 0.80 |
| WET (5) | 1.27 | 1.25 | 1.38 | 1.38 | 4.14 | 1.72 | 1.28 | 1.13 | 1.91 | 1.14 | **1.10** | 2.20 | 2.29 | 1.65 | 2.01 |
| Overall (107) | 1.11 | 0.80 | 0.91 | 1.41 | 4.86 | 1.34 | 0.75 | 0.57 | 1.16 | 0.60 | **0.56** | 1.16 | 1.14 | 1.12 | 1.11 |

CRO=cropland; DBF=Deciduous Broadleaf Forest; EBF=Evergreen Broadleaf Forest; ENF=Evergreen Needleleaf Forest;

MF=Mixed Forest; CSH=Closed Shrubland; WSA=Woody Savanna; SAV=Savanna; OSH=Open Shrubland;

GRA=Grasslands; WET=Wetlands.





**Table 6. Mean bias (in mm day$^{-1}$) for the $E_p$ methods per biome. The best performing method per biome is indicated in bold and is underlined. Based on unstressed days only defined using the energy balance criterion. Different colours are used to group biomes into broader ecosystem types (in descending order: croplands, grasslands, forests, savannah ecosystems, wetlands).**

| | Radiation, Temperature, VPD | | | | | Radiation, Temperature | | | Radiation | | | Temperature | | | |
| | Penman-Monteith | | | Penman | | Priestley and Taylor | | | Milly and Dunne | | | Thornthwaite | | Oudin | |
| | Ref. crop | Standard | Biome | Ref. crop | Standard | Ref. crop | Standard | Biome | Ref. crop | Standard | Biome | Standard | Biome | Standard | Biome |
|---|---|---|---|---|---|---|---|---|---|---|---|---|---|---|---|
| CRO (10) | 1.14 | -0.49 | 0.84 | 2.83 | 2.64 | 2.20 | 0.47 | **-0.01** | 1.43 | -0.24 | 0.12 | -0.65 | -0.59 | -1.62 | -0.62 |
| GRA (20) | 2.65 | 0.53 | 1.16 | 4.37 | 1.90 | 3.69 | 1.11 | **0.02** | 2.57 | 0.22 | -0.10 | -0.14 | -0.44 | -0.61 | -0.73 |
| DBF (15) | 0.30 | -0.48 | 0.89 | 1.30 | 2.63 | 1.81 | 0.94 | **-0.06** | 0.74 | -0.13 | -0.15 | -1.94 | -2.03 | -2.44 | -0.71 |
| EBF (9) | 0.70 | **0.04** | 0.95 | 1.39 | 1.74 | 1.39 | 0.79 | 0.16 | 0.79 | 0.17 | -0.13 | -0.83 | -0.27 | -0.53 | -0.36 |
| ENF (26) | 1.28 | 0.45 | 1.23 | 2.03 | 1.04 | 2.06 | 1.17 | -0.05 | 1.88 | 0.90 | **0.02** | -0.15 | -0.05 | -0.73 | -0.54 |
| MF (4) | 2.22 | 0.65 | 0.30 | 3.31 | 2.04 | 3.26 | 1.46 | -0.07 | 2.53 | 0.87 | -0.04 | 0.73 | 0.19 | **-0.01** | -0.99 |
| CSH (2) | 1.01 | 0.49 | **0.00** | 1.61 | 1.79 | 1.46 | 1.10 | -0.04 | 0.92 | 0.51 | -0.14 | 0.18 | 0.39 | 0.14 | -0.56 |
| WSA (5) | 2.67 | 1.16 | 0.17 | 3.68 | 3.88 | 3.63 | 1.42 | **-0.03** | 2.33 | 0.40 | -0.22 | -0.14 | -0.23 | -0.20 | -0.39 |
| SAV (6) | 2.56 | 1.30 | 0.31 | 3.57 | 3.78 | 3.34 | 1.47 | -0.15 | 2.21 | 0.54 | -0.13 | 0.03 | **0.00** | 0.40 | -0.93 |
| OSH (5) | 4.32 | 1.68 | 0.37 | 6.20 | 2.73 | 5.08 | 2.00 | 0.10 | 3.89 | 1.15 | **0.00** | 1.13 | 0.84 | 0.81 | -0.44 |
| WET (5) | 2.34 | 1.28 | 1.74 | 4.17 | 4.51 | 3.45 | 2.00 | 1.04 | 3.29 | 1.43 | 1.12 | 1.42 | 0.29 | **-0.52** | -2.79 |
| Overall (107) | 1.69 | 0.40 | 0.93 | 2.88 | 2.21 | 2.67 | 1.14 | 0.04 | 1.92 | 0.45 | **0.00** | -0.38 | -0.45 | -0.80 | -0.72 |

CRO=cropland; DBF=Deciduous Broadleaf Forest; EBF=Evergreen Broadleaf Forest; ENF=Evergreen Needleleaf Forest; MF=Mixed Forest; CSH=Closed Shrubland; WSA=Woody Savanna; SAV=Savanna; OSH=Open Shrubland; GRA=Grasslands; WET=Wetlands.





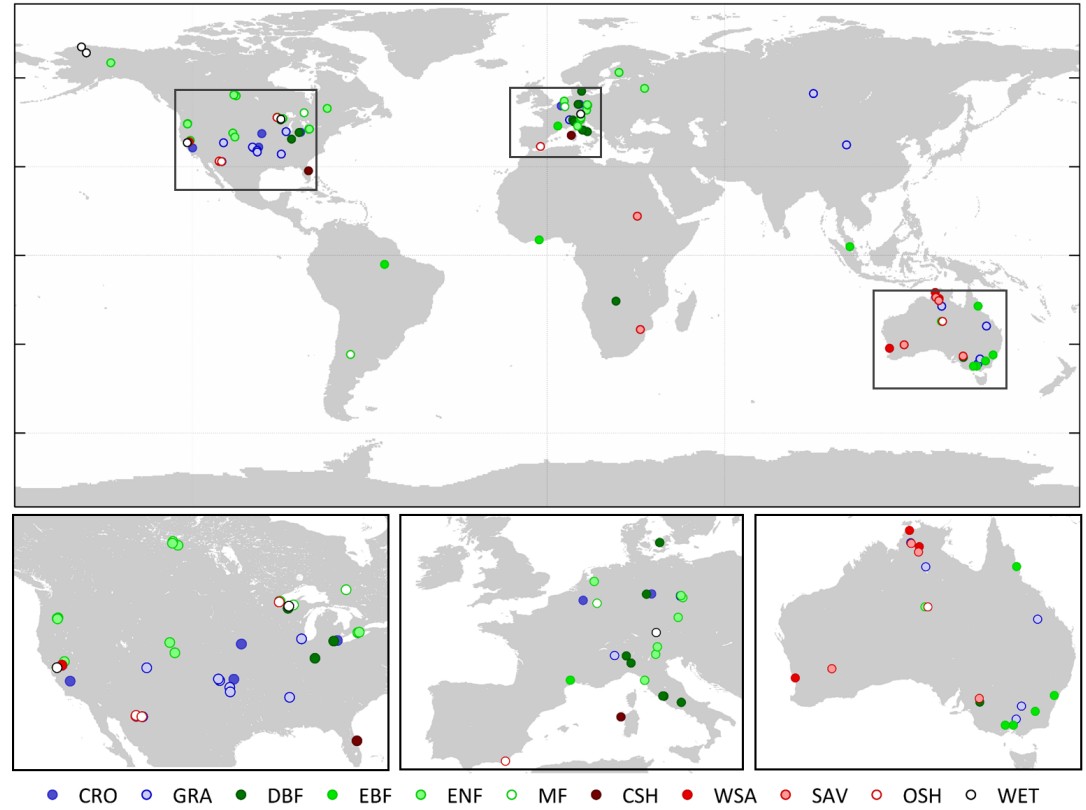

**Figure 1. Location of the flux sites used in this study per biome. CRO=cropland; DBF=Deciduous Broadleaf Forest; EBF=Evergreen Broadleaf Forest; ENF=Evergreen Needleleaf Forest; MF=Mixed Forest; CSH=Closed Shrubland; WSA=Woody Savannah; SAV=Savannah; OSH=Open Shrubland; GRA=Grasslands; WET=Wetlands.**





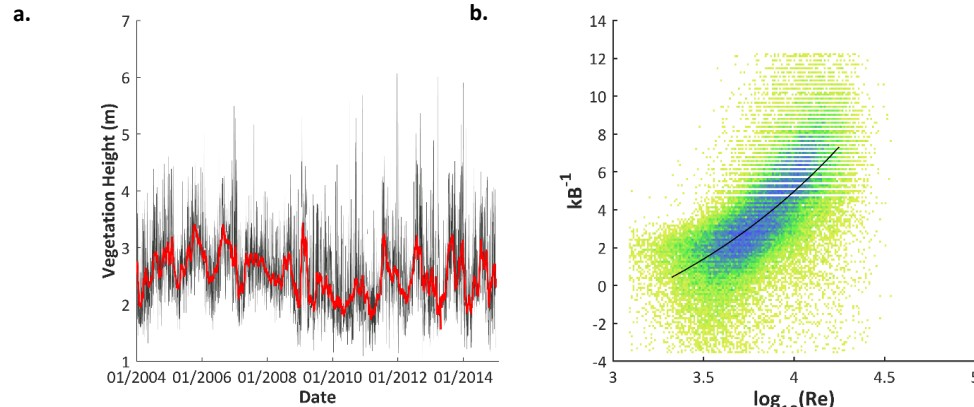

Figure 2. (a) Vegetation height dynamics in time (grey dots: half-hourly measurements; dark grey lines: daily mean vegetation height; red line: 30-day moving average, i.e. the final vegetation height dataset). (b) Relation between Stanton number ($k$B$^{-1}$) and Reynolds number (Re). Both plots correspond to the woody savannah site of Santa Rita Mesquite, US-SRM (Arizona, USA).





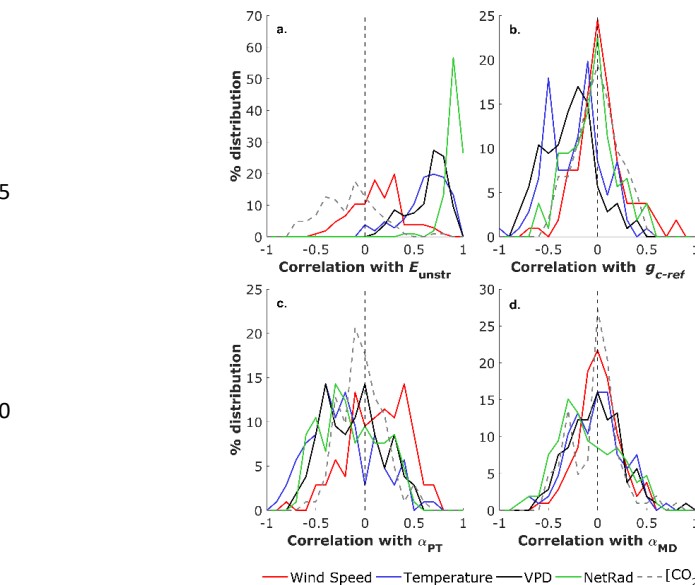

15  **Figure 3. Histograms of correlations between the climate forcing variables and selected key parameters (a)** $E_{unstr}$ **(b)** $g_{c\_ref}$**, (c)** $\alpha_{PT}$
**and (d)** $\alpha_{MD}$ **measured in all flux tower sites. Based on unstressed days only defined using the energy balance criterion.**

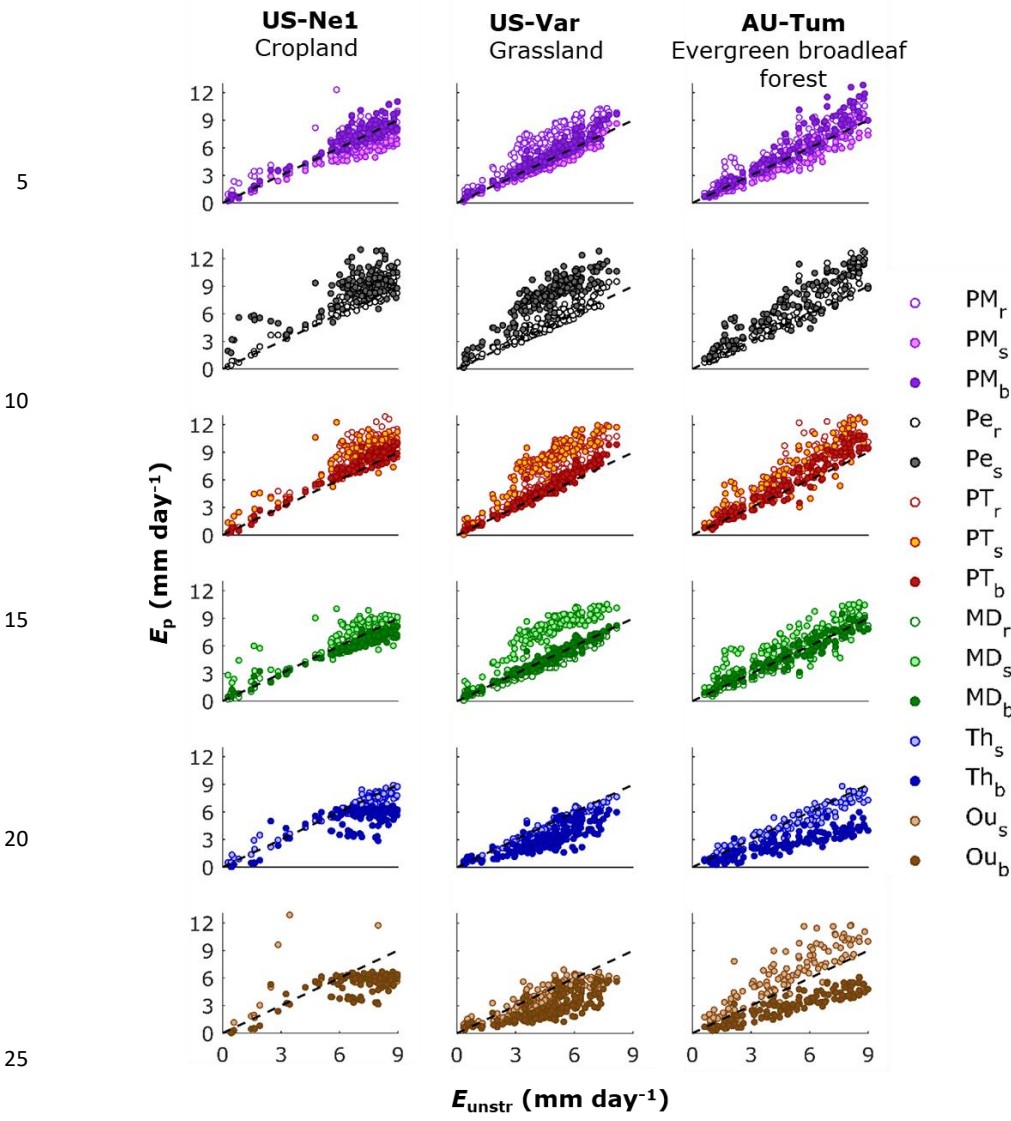

**Figure 4.** Scatterplot of the measured $E_{unstr}$ versus $E_p$ calculated with the different methods. The discontinuous line is the 1:1 line. Based on unstressed days only defined using the energy balance criterion.





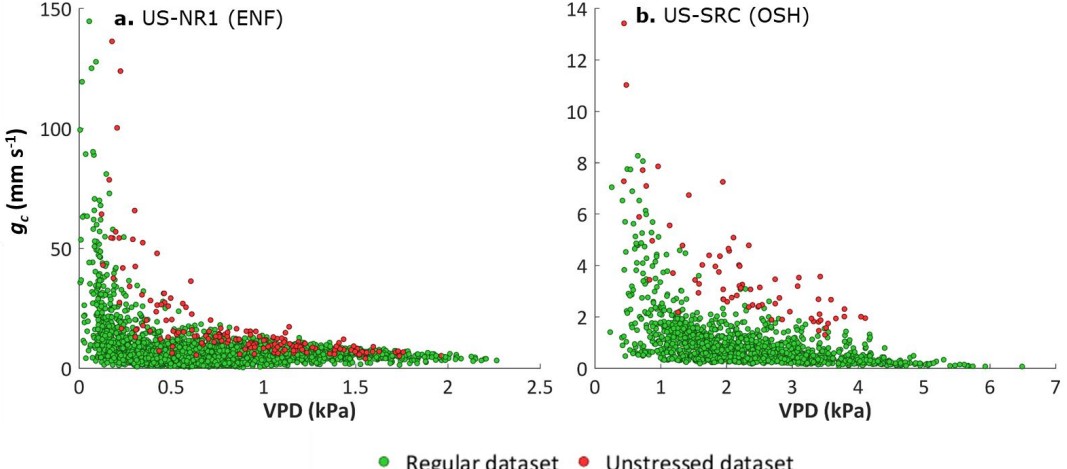

**Figure 5. Canopy conductance** $g_c$ **as a function of vapour pressure deficit (VPD) of the regular and the unstressed datasets of two flux sites, (a) the evergreen needle forest Niwot Ridge Forest and (b) the open savannah woodland site Santa Rita Creosote.**





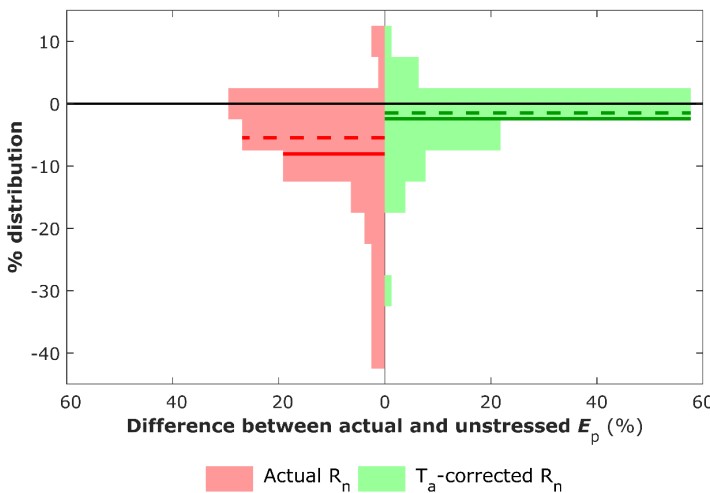

**Figure 6. Comparison of the $E_p$ calculated with a modelled method calculating ($R_n$-G) for unstressed conditions (Section S2) using flux tower data, and the actually observed ($R_n$-G) (red) or a simplified correction of ($R_n$-G) using $T_a$ (Eq. (16)) (green). Negative Y-values indicate a lower estimation of $E_p$ compared to the modelled method. For each distribution, the mean and median are indicated with a full and dashed line, respectively.**





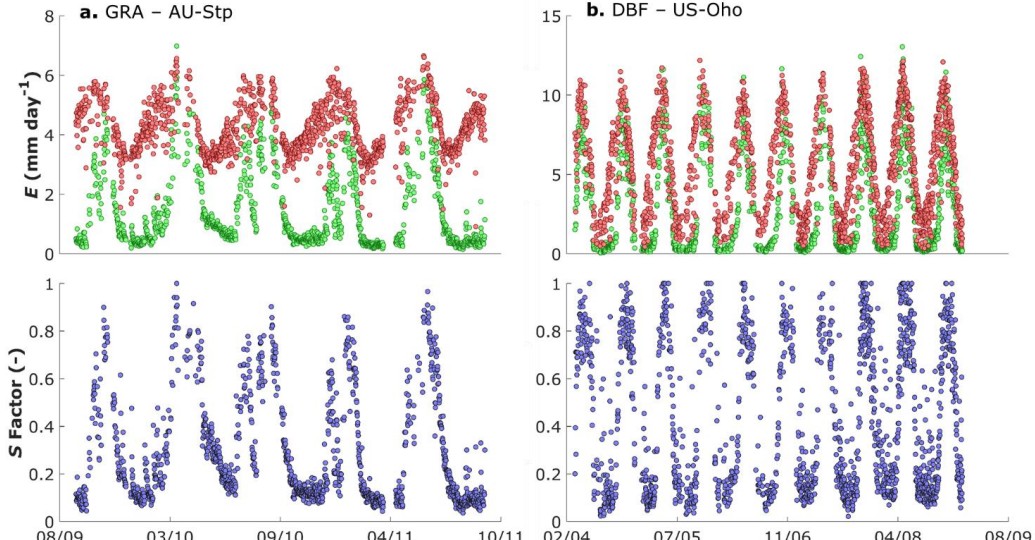

**Figure 7. Seasonal evolution of $E_a$ (top, green), $E_P$ (top, red) and the $S$ factor ($S = E_a E_P^{-1}$, confined between 0 and 1) for two ecosystems, (a) a grassland crop, Sturt Plains in Australia, and (b) a deciduous broadleaf forest, Ohio Oak forest in the USA. $E_P$ was calculated with the $MD_b$-method and using the tower-based correction of ($R_n$-G) as presented in S2 of the supporting information.**