# Peer review of "Potential evaporation at eddy-covariance sites across the globe"

_Hydrology and Earth System Sciences, 2018_

## Referee Comment (RC1) · Ravazzani (Referee) · 26 Nov 2018

This paper presents an analysis of methods to assess potential evaporation and transpiration using data across the globe coming from the FLUXNET database. This is the first time I read this paper even though authors mention the existence of an earlier version of the manuscript in the acknowledgement section. Probably due also to this fact, I found this paper very interesting and well written. My only concern is about the choice of methods to compute evaporation. They present analysis results for methods based on radiation and temperature, methods based on radiation, methods based on temperature. Surprisingly, among these latter, the Hargreaves-Samani method is not included. To my knowledge the Hargreaves-Samani method is widely used and has given satisfactory results in several biomes. So my question is how the methods to

assess evaporation have been chosen and why Hargreaves-Samani equation is not included.

---

## Referee Comment (RC2) · Anonymous Referee #2 · 6 Dec 2018

Performances of different methods widely applied for potential evaporation (Ep) estimation are evaluated using eddy-covariance measurements from the FLUXNET2015 database as reference. A total of 11 biomes types were considered for biome-specific estimation methods. For each eddy-covariance site, performances of the suite of methods are evaluated considering only those days for which ecosystem were classified as unstressed according to certain criteria. I found the paper very well written. All methods and authors' assumptions are well documented and, to my knowledge, fair. After a pleasant reading of the manuscript I made my opinion that the paper can be published as it is (except maybe, the reasonable suggestion by the first reviewer, which I read after making my judgment).

[Figure]

470, 2018.

---

## Author Comment (AC2) · 13 Dec 2018

Dear referee,

Thank you very much for your appreciation of our work. We will include the Hargreave-Samani method in an updated version of the manuscript; please see the response to comments of Referee 1 for a detailed description of the results hereof.

Kind regards,

Wouter Maes On behalf of the co-authors
* * *

---

## Author Response (AR1)

Dear dr. Guadagnini,

Thank you for the positive feedback. We hereby uploaded an updated version of the manuscript, as well as a version highlighting all changes to the manuscript.

As promised earlier, we have incorporated the suggestion by dr. Giovanni Ravazzani to include the Hargreaves-Samani method, both in the text (see also detailed answer to comment by dr. Ravazzani below) and in the tables and figures, as well as in the supporting information (Matlab script).

In addition, we have welcomed the opportunity you provided us to further improve the text. Changes were made throughout the manuscript to improve the readability of the text. Care was taken that the main message of introduction, discussion, or conclusion was not altered and that none of adjustments meeting suggestions by referees of the earlier version of the manuscript were affected.

Thank you once again for considering the manuscript for publication in HESS.

Kind regards,

Wouter Maes,

On behalf of the co-authors.

**Response to comments by RC1 (dr. Giovanni Ravazzani)**
*This paper presents an analysis of methods to assess potential evaporation and transpiration using data across the globe coming from the FLUXNET database. This is the first time I read this paper even though authors mention the existence of an earlier version of the manuscript in the acknowledgement section. Probably due also to this fact, I found this paper very interesting and well written. My only concern is about the choice of methods to compute evaporation. They present analysis results for methods based on radiation and temperature, methods based on radiation, methods based on temperature. Surprisingly, among these latter, the Hargreaves-Samani method is not included. To my knowledge the Hargreaves-Samani method is widely used and has given satisfactory results in several biomes. So my question is how the methods to assess evaporation have been chosen and why Hargreaves-Samani equation is not included.*

Reply:

Dear professor Ravazzani,

First of all, thank you very much for your kind appreciation of our work.

As to the selection of methods, we originally didn't want to include too many methods in the paper, and selected the two temperature-based methods we believed were most successful today – Although the Hargreaves-Samani (HS) method has been used much more than Oudin's method, Oudin's method is very

similar to HS and performed better in Oudin's study (Oudin et al., 2005), and was picked up by several researchers.

However, we understand the comment, and have now redone the analyses to include the HS method. It was calculated as (Oudin et al., 2005; Raziei and Pereira, 2013):

$$\lambda E_p = \alpha_{HS} R_e (T_a + 17.8)\sqrt{T_{max} - T_{min}} \tag{1}$$

With $\alpha_{HS}$ a constant, $T_a$ the daily mean air temperature, $T_{max}$ the daily maximum and $T_{min}$ the daily minimum air temperature. As for the other temperature-based methods, two versions were calculated. In the standard version, $\alpha_{HS}$ =0.0023; in the biome-specific version, $\alpha_{HS}$ was calibrated per biome.

We then re-did all calculations and analyses. The results are summarized in the tables 1-3 (mean correlation, unbiased RMSE and bias for the energy balance criterion) and 4-6 (same variables, but for the soil moisture criterion.

All in all, the HS method performs best of the three temperature-based methods, but clearly does not perform as good as the simple radiation-based (Milly and Dunne) method or the Priestley and Taylor method, and this for both unstressed subset selection criteria. Other analyses (ie S6-S11 in Supplement of original document) were also performed and are in line with these observations. Hence, the overall conclusions of the paper will not be affected by including the Hargreaves-Samani method.

In the revised version of the text, we will include the Hargreaves-Samani method.

Kind regards,

Wouter Maes,

On behalf of the co-authors

[revised manuscript text omitted]
 (10) | 38.3 ± 23.0 | 1.15 ± 0.14$^{a}$ | 0.86 ± 0.09$^{a}$ | 38.7 ± 14.5$^{ab}$ | 77.0 ± 27.8$^{b}$ | 2.96 ± 0.69$^{ab}$ |
| GRA (20) | 30.5 ± 40.2 | 1.02 ± 0.16$^{ab}$ | 0.74 ± 0.12$^{ab}$ | 30.4 ± 13.9$^{b}$ | 103.2 ± 38.9$^{b}$ | 2.32 ± 0.70$^{bc}$ |
| DBF (15) | 32.6 ± 27.4 | 1.09 ± 0.14$^{ab}$ | 0.80 ± 0.08$^{ab}$ | 33.3 ± 7.8$^{b}$ | 70.5 ± 18.0$^{b}$ | 3.39 ± 0.83$^{a}$ |
| EBF (9) | 42.0 ± 36.6 | 1.09 ± 0.15$^{ab}$ | 0.74 ± 0.05$^{abc}$ | 53.1 ± 16.8$^{a}$ | 95.5 ± 22.9$^{b}$ | 3.07 ± 0.57$^{ab}$ |
| ENF (26) | 28.4 ± 52.1 | 0.89 ± 0.26$^{b}$ | 0.62 ± 0.09$^{c}$ | 40.3 ± 16.7$^{ab}$ | 92.0 ± 21.8$^{b}$ | 2.78 ± 0.76$^{ab}$ |
| MF (4) | 10.0 ± 7.1 | 0.88 ± 0.23$^{ab}$ | 0.64 ± 0.13$^{bc}$ | 26.1 ± 3.6$^{b}$ | 138.2 ± 91.6$^{ab}$ | 2.21 ± 0.97$^{abc}$ |
| CSH (2) | 8.5 ± 3.9 | 0.90 ± 0.10$^{ab}$ | 0.64 ± 0.15$^{abc}$ | 41.4 ± 13.7$^{ab}$ | 130.3 ± 36.1$^{ab}$ | 2.03 ± 0.68$^{abc}$ |
| WSA (5) | 8.4 ± 3.4 | 0.95 ± 0.09$^{ab}$ | 0.70 ± 0.10$^{abc}$ | 33.8 ± 6.4$^{ab}$ | 104.6 ± 19.7$^{b}$ | 2.25 ± 0.51$^{abc}$ |
| OSH (5) | 7.8 ± 3.7 | 0.87 ± 0.14$^{b}$ | 0.68 ± 0.15$^{c}$ | 35.0 ± 4.1$^{ab}$ | 147.1 ± 63.9$^{ab}$ | 1.88 ± 0.61$^{c}$ |
| SAV (6) | 4.3 ± 2.0 | 0.79 ± 0.11$^{b}$ | 0.58 ± 0.09$^{bc}$ | 31.3 ± 11.2$^{ab}$ | 147.7 ± 61.8$^{ab}$ | 1.59 ± 0.38$^{bc}$ |
| WET (5) | 20.0 ± 14.1 | 1.03 ± 0.47$^{ab}$ | 0.75 ± 0.11$^{b}$ | 17.8 ± 13.3$^{b}$ | 638.6 ± 1230.1$^{a}$ | 2.00 ± 0.54$^{bc}$ |

CRO=cropland; DBF=Deciduous Broadleaf Forest; EBF=Evergreen Broadleaf Forest; ENF=Evergreen Needleleaf Forest; MF=Mixed Forest; CSH=Closed Shrubland; WSA=Woody Savanna; SAV=Savanna; OSH=Open Shrubland; GRA=Grasslands; WET=Wetlands.

**Table 3. Influence of atmospheric conditions on $E_{\text{unstr}}$ and on selected key parameters ($g_{c\_ref}$, $\alpha_{\text{PT}}$, $\alpha_{\text{MD}}$). (left) Mean ± 1 standard deviation of the correlations of $E_{\text{unstr}}$, $g_c$, $\alpha_{\text{PT}}$ and $\alpha_{\text{MD}}$ against the atmospheric conditions, and (right) number of sites (out of total of 107) with significant negative/positive correlations between $E_{\text{unstr}}$, $\alpha_{\text{PT}}$, $g_{c\_ref}$ and $\alpha_{\text{MD}}$ and the climate forcing variables. Based on unstressed days only defined using the energy balance criterion.**

| | Mean ± 1 standard deviation of the correlations | | | | Number of sites with significant negative/positive correlations | | | |
|---|---|---|---|---|---|---|---|---|
| | $E_{\text{unstr}}$ | $g_{c\_ref}$ | $\alpha_{\text{PT}}$ | $\alpha_{\text{MD}}$ | $E_{\text{unstr}}$ | $g_{c\_ref}$ | $\alpha_{\text{PT}}$ | $\alpha_{\text{MD}}$ |
| Wind | $0.13 \pm 0.26$ | $0.03 \pm 0.25$ | $0.12 \pm 0.31$ | $0.01 \pm 0.22$ | 6/26 | 4/13 | 11/30 | 5/6 |
| $T_{\text{air}}$ | $0.60 \pm 0.24^{*}$ | $-0.22 \pm 0.29$ | $-0.21 \pm 0.34$ | $-0.02 \pm 0.28$ | 0/93 | 43/0 | 43/5 | 16/13 |
| VPD | $0.64 \pm 0.20^{*}$ | $-0.27 \pm 0.27$ | $-0.11 \pm 0.31$ | $-0.01 \pm 0.28$ | 0/93 | 48/0 | 31/10 | 15/11 |
| $R_n$ | $0.90 \pm 0.08^{*}$ | $-0.05 \pm 0.25$ | $-0.13 \pm 0.30$ | $-0.10 \pm 0.31$ | 0/106 | 17/3 | 33/5 | 30/14 |
| $[CO_2]$ | $-0.16 \pm 0.30$ | $-0.01 \pm 0.23$ | $-0.03 \pm 0.22$ | $-0.03 \pm 0.25$ | 34/5 | 7/5 | 9/4 | 12/4 |

*significantly different from 0

[revised manuscript text omitted]